

# Spatial Heterodyne Observations of Water (SHOW) from a high altitude airplane: Characterization, performance and first results

Jeffery Langille[1], Daniel Letros[1], Adam Bourassa[1], Brian Solheim[1], Doug Degenstein[1], Fabien Dupont[2], Daniel Zawada[1]

[1]Institute of Space and Atmospheric Studies, University of Saskatchewan, Saskatoon, S7N 5E2, Canada
[2]ABB Inc., Quebec city, G1P 0B2, Canada

*Correspondence to*: Jeffery Langille (jeff.langille@usask.ca)

**Abstract:** The Spatial Heterodyne Observations of Water instrument (SHOW) is a limb-sounding satellite prototype that utilizes the Spatial Heterodyne Spectroscopy Technique (SHS), operating in a limb-viewing configuration, to observe limb-scattered sunlight in a vibrational band of water vapour within a spectral window from 1363 nm to 1366 nm. The goal is to retrieve high vertical and horizontal resolution measurements of water vapour in the upper troposphere and lower stratosphere. The prototype instrument has been configured for observations from NASA's ER-2 high altitude airborne remote science airplane. Flying at a maximum altitude of ~21.34 km with a maximum speed of ~760 km /hr, the ER-2 provides a stable platform to simulate observations from a low-earth orbit satellite. Demonstration flights were performed from the ER-2 during an observation campaign from July 15 – July 22, 2017. In this paper, we present the laboratory characterization work and the Level 0 to Level 1 processing of flight data that was obtained during an engineering flight performed on July 18th, 2017. Water vapour profile retrievals are presented and compared to in situ radiosonde measurements made of the same approximate column of air. These measurements are used to validate the SHOW measurement concept and examine the sensitivity of the technique.

## 1    Introduction

Water vapour is an extremely important trace species in the upper troposphere and lower stratosphere (UTLS) region of Earth's atmosphere. Indeed, it is well known that the abundance and distribution of water vapour in the UTLS is strongly linked to climate processes (Gettleman et al., 2011; Sherwood et al., 2010). However, research over the past couple decades has indicated that the distribution of water vapour and its link to climate processes is not fully understood. For example, the impact of Stratosphere Troposphere Exchange (STE) and the formation of the Tropopause Inversion layer (TIL), as well as, the role of water vapour as mechanism for radiative feedback still requires detailed study (Randel and Jensen, 2010). The primary factor limiting the ability to perform a detailed study of these processes is the lack of accurate long term global measurements that have a high vertical resolution in the UTLS.

Remote sensing of water vapour can be achieved using many different techniques and platforms (space-based, balloon, in-situ, airplane or ground-based platforms). While each technique has its advantages, the best combination of vertical resolution and global coverage of trace species in the UTLS is achieved using a limb-viewing instrument operating from a low-earth orbit satellite. Indeed, the current measurement record of water vapour has been enriched with observations from limb viewing instruments such as the Michelson Interferometer for Passive Atmospheric Sounding (MIPAS) (Fischer et al., 2008; Milz et al., 2005; Stiller et al., 2009), the Microwave Limb Sounder (MLS) (Hurst et al., 2014; Hurst et al., 2016; Sun et al., 2016), and the Scanning Imaging Absorption Spectrometer for Atmospheric CHartographY (SCIAMACHY) (Rozanov et al., 2011). These instruments provide between 3 – 5 km vertical resolutions in the UTLS with < ±1 ppm error.



The Spatial Heterodyne Measurements of Water (SHOW) instrument is a prototype satellite instrument that is being developed in collaboration between the Canadian Space Agency (CSA), the University of Saskatchewan (USASK) and ABB Inc, to provide high spatial resolution measurements (< 500 m) of the vertical distribution of water vapour in the UTLS from a space borne platform (Langille et al., 2017; Langille et al., 2018). The instrument utilizes the Spatial Heterodyne Spectroscopy (SHS) technique (Connes, 1958; Harlander, 1991), operating in the limb-viewing configuration to observe limb scattered radiation within a vibrational band of water centered at 1364.5 nm. The prototype version of the instrument, discussed in this paper, is configured for measurements from NASA's ER2 high - altitude airborne science airplane. High altitude measurements (~21 km) from the ER-2 airplane provide a sub-orbital demonstration of the SHOW instrument technique. A vertically resolved image of the limb scattered water vapour absorption spectrum with an un-apodized spectral resolution of ~0.03 nm within a narrow spectral window from 1363 nm and 1366 nm is extracted from each limb image. These vertically resolved spectra are inverted using non-linear inversion techniques and a forward model of the measurement to retrieve the vertical distribution of water vapour.

The SHOW measurement approach is unique in several ways. For example, it is the first demonstration of the measurement of atmospheric water vapour using the SHS technique; specifically, using limb scattered sunlight. In addition, the majority of the SHS instruments that have been developed, have been used to observe well-isolated emission features, such as in the case of SHIMMER (Harlander et al., 2002), and the DASH type instruments that are used to remotely detect motions in the upper atmosphere using well isolated airglow emissions (Englert et al., 2016; Englert et al., 2015). Therefore, SHOW is also one of only several SHS instruments (Englert et al., 2009) that has been developed to observe absorption features over a broader spectral range.

In two earlier publications, we presented the design of the prototype instrument (Langille et al, 2017) and demonstrated that the configuration and sensitivity of the instrument was suitable for airplane measurements of water vapour (Langille et al., 2018). This sensitivity study was performed assuming an ideal instrument configuration with realistic signal levels and Poisson noise added to the signals. A non-linear optimal estimation retrieval algorithm was developed to invert the spectral signatures and retrieve water vapour. Assuming the ideal configuration and an airplane altitude of 22 km, it was shown that SHOW is capable of providing vertically resolved measurements of water vapour with 1-ppm accuracy with 500 m – 1 km vertical resolution in the 8 km – 18 km altitude range.

In this paper, we focus on the Level 0 to Level 1 processing and characterization of the SHOW measurements that were obtained from NASA's ER-2 airplane during an engineering flight performed on July 18, 2017. During the flight, a radiosonde was launched from the Jet Propulsion Laboratory (JPL) facility located close to Table Mountain at the same time that SHOW observed the same approximate column of air. This provides a direct estimate of the in-situ water vapour abundance for the coincident SHOW measurements.

In practice, it was found that non-ideal instrument effects associated with the instrument configuration introduced systematic variations in the spectra that must be appropriately characterized prior to performing water vapour retrievals. In this paper, we present the work that was performed to characterize these effects. It is shown that knowledge of the instrument configuration can be utilized to develop an instrument model that is optimized to predict the systematic variations that are observed in the SHOW spectra. The approach is validated by using the measured in-situ water vapour abundance as input to a forward model to simulate the expected limb-scattered radiance profile for the coincident measurements. This radiance profile is then used as input to the SHOW instrument model to simulate the expected SHOW interferograms and spectra. The ability of the model to predict systematic variability in the Level 1 spectra is examined. Preliminary water vapour retrievals are presented and compared to the in-situ



measurements. These measurements are used to demonstrate the SHOW measurement technique and examine the performance of the instrument.

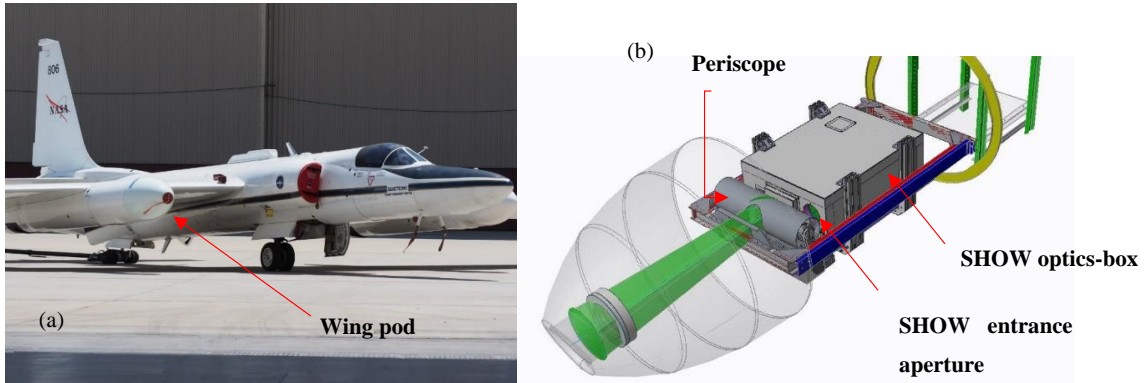

**Figure 1 The NASA ER-2 airplane (a) and a schematic of the SHOW instrument depicted inside the ER-2 wing pod (b)**

## 2      Spatial Heterodyne Observations of Water (SHOW) from NASA's ER-2 high - altitude airborne science airplane

The primary scientific goal of the SHOW instrument concept is the realization of high vertical resolution sampling (< 500 m) of the water vapour distribution in the UTLS region from a low earth orbit satellite with < ±1 ppm accuracy. The SHOW instrument was originally developed as a laboratory prototype under the CSA Advanced Studies Program at York University and the instrument was further adapted through CSA's Space Technology Development Program (STDP) for demonstration from stratospheric balloon or high altitude airplane. The instrument was tested from a stratospheric balloon in 2014 (Dupont et al., 2014) and from 2015 to 2016, the instrument was configured for observations from NASA's ER-airplane (Dupont et al., 2016; Bourassa et al., 2016).

The SHOW prototype instrument is shown mounted inside the wing pod of ER-2 airplane in Figure 1. The instrument observes the limb through a forward-looking window that is located at the front of the wing pod (see Figure 1 (a)). Limb-scattered sunlight enters the instrument through a periscope assembly that is mounted at the input of the optics box. This assembly consists two mirrors that align the optical axis of the instrument with the wing pod window as shown in Figure 1(b). The aft optics are configured to provide a 4º vertical by 5.1° horizontal field of view of the limb and the instrument is tilted inside the wing pod to point the SHOW field of view downward by 3.23 degrees. The relative tilt between the airplane boresight and the wingpod combines to tilt the optical axis of the instrument downward 2.40 degrees from the airplane boresight.

The vertical sampling at the limb is defined by the airplane viewing geometry from the ER-2, and the configuration of the optical system and the SHS. In general, an SHS produces an interferogram image that corresponds to a set of overlapping Fizeau fringes with spatial frequencies that depend on the separation of the signal wavelength from the heterodyne (Littrow) wavelength. For SHOW, the SHS and aft optics are arranged so the interferograms are aligned horizontally. The fore-optics is configured to be anamorphic so the interferograms are imaged conjugate to the limb in the vertical dimension at the detector, whereas, each interferogram sample in the horizontal averages over the horizontal scene information. This ensures that the spatial information contained in the horizontal does not contaminate the spectral information recorded at a particular row.

Each SHOW measurement provides an interferogram image that has 295 vertical rows and 494 interferogram samples. The geographic location of the range of tangent altitudes within the field of view are determined from the airplane attitude information



(altitude, heading, pitch, roll etc.). Variations in the pitch of the airplane during flight result in different interferogram rows observing different tangent altitudes during different parts of the flight. Post flight, the airplane attitude information is used to obtain a one-to-one mapping of each interferogram row to tangent altitude. For the measurements presented in this paper, the lowest observed tangent (set by the pitch of the airplane) varies from ~1 km to ~7km. Assuming zero pitch at an airplane altitude

of 21.34 km and taking the radius of the Earth to be 6371 km, the SHOW viewing geometry (4 degrees vertical, tilted 2.40 degrees downward from the boresight) provides an altitude coverage from 2.5 km up to 21 km. Within this altitude range, SHOW obtains 295 vertical samples with an instantaneous angular resolution of 0.0176 degrees. This corresponds to an approximate tangent point vertical resolution of 20 m – 151 m, where the sampling resolution decreases with increasing altitude from ~2.5 km up to ~21 km.

During Level 0 to Level 1 processing, the raw interferogram signals are converted to calibrated interferograms and these calibrated

interferograms are converted into corrected spectra. For each row in the image, a water vapour spectrum is obtained with an un-apodized spectral resolution of ~0.03 nm. A single processed SHOW data record includes the raw interferogram image, a vertically resolved image of the water vapour spectrum, as well as, housekeeping information and a mapping of each row in the image to tangent altitude at the limb. The SHOW retrieval approach, described in (Langille et al., 2018), implements a non-linear optimal estimation algorithm (Rodgers, 2000) to extract the vertical distribution of water vapour from the geo-referenced, vertically

resolved water vapour spectra. The primary specifications of the SHOW prototype instrument are listed in Table 1. The reader is referred to (Langille et al., 2017; Langille et al., 2018) for more details regarding the design of the prototype instrument.

| Instrument parameter | Specification |
|---|---|
| **SHOW ER-2 altitude** | **~21.34 km (70000 ft max)** |
| **Airplane speed** | **~760 km/hr (maximum at altitude)** |
| **Field Of View** | **4º vertical by 5.1º horizontal** |
| **Instantaneous angular vertical resolution** | **0.0176 degrees** |
| **Mass** | **222.68 lbs [101 kg]** |
| **Power** | **465 W (peak), 200 W (average)** |
| **Dimensions** | **(0.465 m × 1.32 m× 0.38 m )** |
| **Operating Temperature** *SHS temperature* | **-90°C (at 60 kft altitude) to +40°C +23°C** |
| **Spectral Resolution (unapodized)** | **~0.03 nm** |
| **Spectral range** | **1363 nm – 1366 nm** |

**Table 1 SHOW ER-2 instrument parameters**

### 3      Data acquisition and storage

Control and operation of the instrument from the ground is achieved using the flight control software developed by ABB Inc and ground support equipment developed by USASK. Communication with the instrument is performed using the NASA Airborne Science Data and Telemetry system (NASDAT) and the airplane attitude information is also monitored and stored onboard the



airplane. The control software and communication system allows for near real time download of images, as well as, the airplane attitude information (altitude, speed, pitch, roll, heading etc.) during the flight. The system was used for configuring the science modes (acquisition times, frame rates, FPA temperature etc.) and to monitor housekeeping data that is vital to the instrument survival; such as, PCB temperatures and SHS temperatures.

A 29 GB data storage unit is used to store the SHOW measurements onboard the instrument. The SHOW detector array is a Raptor OWL 640 camera with 512 x 640 15 um InGaAs pixels. Each acquired image is 650 kB and is stored along with the SHOW housekeeping data. For the ER-2 measurements, we use two primary modes of operation - a 1 HZ sampling mode with a 900 ms integration time and 0.5 Hz sampling mode that uses 1800 ms integration time. The two configurations are used in the case of higher and lower signals that correspond to forward scattering and back scattering signals respectively. Assuming the 1 Hz frame

rate, the data storage capacity is met in approximately 12 hours.

Post flight, the raw data is downloaded and processed to generate a Level 0 netCDF file for each specific science mode chosen during the flight. The netCDF file stores the raw image files along with the measurement configuration, such as integration time, image size, UTC time and housekeeping data. The corresponding airplane attitude information is stored in a separate data file provided by NASA. The raw data is processed using the Level 0 to Level 1 processing chain to produce Level 1 calibrated

interferograms and Level 1 spectra. The Level 1 data is stored as a netCDF file with each data record containing one minute of data.

## 4     Thermal Control

An important aspect of the design of the SHOW prototype is the thermal stability of the instrument. The two most temperature sensitive components are the narrowband filter and the SHS, both of which were designed for operation near room temperatures.

Since the SHS is not thermally compensated, the heterodyne wavelength of the SHS shifts slightly with temperature. For the SHOW instrument this shift occurs at a rate of approximately ~0.06 nm per degree Celsius. On the other hand, the narrowband filter drifts at a rate of ~0.02 nm per degree Celsius. In practice, the shift in Littrow can be tracked during the flight by lining up (or fitting) the spectrum to the known water vapour absorption features. In any case, it is important to actively control the temperature of the SHOW optics box in order to minimize the extent of the thermal drift.

In the worst case, the inside of the wing pod will reach the ambient temperature where the extreme temperatures can easily reach 40 degrees Celsius on the tarmac and –90 degrees Celsius during flight. Therefore, the SHOW optics are housed within a thermally controlled environment (optics-box) designed and built by ABB Inc that can be operated in near vacuum conditions in the ambient temperature range from -90 degrees Celsius to 40 degrees Celsius. The temperature of the top and bottom plates of the optics-box are actively controlled using resistive heating elements. On the other hand, the SHOW detector is cooled to 0 degrees Celsius using

a thermo-electric cooler. Heat generated by the detector is dissipated by thermally strapping the detector to the frame of the outer enclosure of the box, which is thermally connected to the frame of the wing-pod. Prior to flight, the instrument is purged with Nitrogen for over 24 hours to remove moisture and to achieve a stable operating environment. The active thermal control of the optics-box is designed so that the SHS temperature stabilizes at approximately 23 degrees Celsius. The temperature of the optics-box, the wing-pod window, as well as, the SHOW periscope are monitored during the flight.




## 5    SHS spectroscopy

As noted in the introduction, non-ideal instrument effects associated with the instrument configuration result in systematic variations in the observed spectra that need to be characterized to perform water vapour retrievals. In order to understand and interpret these variations, it is important to review some key concepts regarding the SHS technique. The most rigorous theoretical

background of SHS is provided by Harlander (1990) and Harlander (1991). Some of the more recent applications of the SHS technique for the remote sensing of the atmosphere can be found in (Englert, et al., 2015). The framework and notation that is used here to describe the SHS technique follows directly from work that was performed at the US naval laboratory (Englert et al., 2004; Englert et al., 2006).

Conceptually, the SHOW instrument operates in a similar manner to a traditional limb imaging FTS instrument such as MIPAS

(Fischer et al., 2008); however, in this case, a limb image of the water vapour absorption spectrum is obtained from each frame without vertical scanning and without moving parts in the interferometer. The basic configuration of an SHS is similar a Michelson interferometer that has its plane mirrors replaced by fixed, tilted diffraction gratings. Field-widening of the instrument is achieved by placing appropriately selected prisms in the arms of the interferometer. Collimated light enters the spectrometer and is incident on a beamsplitter that transmits and reflects (respectively) 50% of the input radiation down the two equal length arms of the

interferometer. At the end of each arm, the radiation is incident on a diffraction grating that is tilted by the Littrow angle $\theta_L$. In this configuration, a wavenumber dependent shear $\gamma(\sigma)$ is introduced between the wavefronts that exit the interferometer. An image of the associated wavenumber dependent Fizeau fringes are formed by imaging the grating onto a detector. The technique provides a heterodyned interferogram about the Littrow wavelength with a spatial frequency on the detector given by:

$\kappa = 2\sigma sin(\gamma) \approx 4(\sigma - \sigma_0) tan\,\theta_L$                                                                            (1)

, where the approximation assumes small $\gamma$, and $\sigma_0$ is the Littrow wavenumber for which $\gamma = 0$. Therefore, the spacing of the Fizeau fringes depends on the wavelength difference from the central heterodyne wavelength (or Littrow wavelength) and the zero spatial frequency fringe is generated by incident radiation at the Littrow wavelength.

For the SHS, each pixel essentially behaves as a separate detector, making the system extremely sensitive to variations in the intensity across the interferogram. Non-ideal flat field calibrations, bad/dead/hot pixels, and pixel-to-pixel non-uniformities (etc.) result in systematic variability in the interferograms and subsequently, errors in the retrieved spectra. The presence of phase distortions in the interferograms can also be important. In general, the basic form of an SHS interferogram, which includes these effects, can be described [see Englert et al., 2006] using Eq.2.

$I(x,y) = \int_0^\infty B(\kappa)\tau(\kappa)\{t_A^2(x,y) + t_B^2(x,y) + 2\varepsilon(x,\kappa)t_A(x,y)t_B(x,y) \cdot cos[2\pi\kappa x + \Theta(x,y,\kappa)]\}d\kappa$                    (2)

In this equation, the subscripts A and B denote the two arms of the interferometer, $B(\kappa)$ is the spectral radiance, $\tau(\kappa)$ is the filter transmission, $\varepsilon(x,\kappa)$ is the modulation efficiency of the interferometer, $t_A(x,y)$ and $t_B(x,y)$ are the transmission coefficients for

each arm respectively and $\Theta(x,y,\kappa)$ is a phase distortion term. We have also substituted $\kappa = 4(\sigma - \sigma_o)tan(\theta_L)$ where $\sigma$ is the



wavenumber and $\sigma_o$ and $\theta_L$ are the Littrow wavenumber and Littrow angle respectively. For the SHOW instrument, each vertical row in the image is mapped one-to-one to a particular tangent altitude at the limb. Therefore, the interferogram equation is written with explicit dependence on the x and y position.

A robust theoretical framework has been developed by (Englert et al., 2004; Englert et al., 2006) that provides a template to correct the interferogram described by Eq.2. This includes the flat field correction and phase distortion. Once the flat field correction is performed, the corrected interferogram can be written as, Eq. 3 where we have substituted $B'(x, \kappa) = B(\kappa)\tau(\kappa)\varepsilon(x, \kappa)$.

$$Ic = \int_0^\infty B'(x,\kappa)cos[2\pi\kappa x + \Theta(x,y,\kappa)]\,d\kappa \qquad (3)$$

If $\Theta(x,y,\kappa) = 0$, the corrected interferogram has the familiar form of a typical FTS interferogram where $\kappa$ has replaced $\sigma$ as the variable. For an FTS, we strictly have $\sigma > 0$ and a simple FFT of the interferogram returns the spectrum (Davis et al., 2001). On the other hand, if $\Theta(x,y,\kappa) \neq 0$, phase errors are introduced and a complex spectrum is returned by the FFT. In general, the phase errors are corrected using phase correction algorithms (Revercomb et al., 2003; Davis et al, 2001). For the SHS, one can perform an analogous correction, called a phase distortion correction, using the technique developed by (Englert et al., 2006). For

the SHOW instrument, the ability to perform this correction is limited by aliasing effects, which we now describe.

Observe that an SHS cannot distinguish between signals that are generated by wavenumbers separated by the same amount on either side of Littrow ($\pm\kappa$). This results in aliasing of spectral information from one side of Littrow to the other. Generally, this effect is minimized by choosing a narrow band filter such that the Littrow wavenumber is close to the edge of the passband so we have: $\kappa > 0$. In this case, we can symmetrize Eq.3, the simple FFT recovers the spectrum and a phase distortion correction is

feasible. The SHOW filter and SHS were both designed to operate at 20C in the laboratory and there was minimal aliasing at 20C (Langille et al., 2018), however, the ER-2 environment required an operating temperature of 23C and we did not have time or the financial resources to change the filter for the first flight. Consequently, signals with $\kappa < 0$ are transmitted through the passband. In our earlier study, we showed that the impact of these aliased signals on the SHOW retrievals associated with this effect is minimal when the phase term in Eq.2 is negligible (*i.e when* $\Theta(x,y,\kappa)\sim 0$).

In the prototype version of the SHOW instrument, the aliasing effect is complicated by the presence of a linear phase variation: $\Theta(\kappa,y) = 2\pi\left[\frac{\kappa}{4\tan(\theta_L)} + \sigma_o\right]y\alpha$ in the y direction. This phase variation was introduced by design, to allow for the characterization of the Littrow wavelength in the laboratory using well-isolated monochromatic calibration lines (see Section 9.5). The phase variation is generated by introducing a slight tilt $\alpha$ between the two gratings in the y-direction which has the effect that the Fizeau fringes associated with signals from either side of Littrow are rotated in opposite direction. In the case of a monochromatic line,

once the side of Littrow has been identified, the location of Littrow can be experimentally determined using Eq.1 from knowledge of the observed spatial frequency and knowledge of the known calibration line position.

On the other hand, taking the FFT in the presence of the linear phase term, $\Theta(\sigma,y)$ in Eq.3 returns a complex spectrum. Aliasing in the SHS, signals with ($\kappa < 0$) exist and introduce additional signals to the complex spectrum that do not exist in the context of the traditional FTS. These signals combine to form an "effective" complex spectrum that contains systematic variations which are

difficult to isolate analytically. The best that one can do is directly measure the phase term $\Theta(\kappa,y)$ in the laboratory using a diode laser following the approach discussed by (Englert et al., 2004). In addition, the functional shape of the effective spectrum can be





determined by taking measurements of a white light source in a vacuum to remove contamination from water vapour absorption features. Unfortunately, for the SHOW characterization work, we did not have access to a large enough vacuum tank to accommodate the instrument or a laser diode to perform the phase characterization.

In this paper, we attempt to use an instrument model to predict this behavior with the assumption that the phase term is linear and

5 given by: $\Theta(\kappa, y) = 2\pi \left[ \frac{\kappa}{4\tan(\theta_L)} + \sigma_o \right] y\alpha$. This instrument model is optimized using knowledge of the instrument configuration that is obtained through laboratory characterization work, as well as, flight measurements that are obtained while observing a column of air that is simultaneously sampled using a radiosonde to measure the in-situ water vapour abundance. This approach has several limitations. For example, the ability to adequately capture the systematic variations is limited by our knowledge of the true phase variation, as well as, the knowledge of the atmospheric state. In this paper, we attempt to isolate and characterize the

10 remaining systematic errors and we examine the impact on the retrieval of water vapour with SHOW.

## 6  Noise

The primary source of noise in the SHOW interferograms is associated with counting statistics and the inter-pixel variability of the detector response. Poisson statistics describe the measurement of the incident photon signal from the atmosphere (S), the thermally generated dark signal (D) and the readout noise (R). On the other hand, inter-pixel variations associated with incomplete flat field

15 corrections (optical variations due to dust, aberrations etc.) and variations in the relative pixel response (known as photo-response non-uniformity (PRNU))(Ferrero et al., 2006; Schulz et al., 1995) across the detector introduce an inter-pixel variability (spatial noise) to the samples in the interferograms. This variability introduces uncertainty in the interferogram samples that is not described by photon counting statistics.

Generally speaking, flat field calibrations are performed to remove the intensity variations associated with optical effects and a

20 non-uniformity correction is performed to minimize the impact of the relative pixel response. Ideally, the relative pixel response is corrected separately from the flat field variations by characterizing the PRNU prior to installing the camera in the instrument. For the SHOW prototype, this was not possible since the camera was installed by ABB prior to shipping the instrument for the calibration work. In this case, the flat field correction is performed in combination with the non-uniformity correction by obtaining white light flat field measurements at signal levels that closely match signal levels recorded during flight. The PRNU is

25 characterized in the laboratory by performing a two-point non-uniformity correction using white light flat field measurements.

The SHOW flat field calibrations are performed using the flat field approach developed by (Englert et al., 2006) and described in Section 9.4. After applying the flat field corrections, the remaining inter-pixel variations are associated with the residual photo-response non-uniformity of the detector. Techniques have been developed to characterize these variations in the interferogram images in the laboratory, as well as, their influence on the spectral measurements. In the interferogram dimension, these variations

30 add a relative error to the signal at each sample. Assuming a random relative pixel variation $\frac{\delta S}{S}$, this results in a SNR in the spectral measurements of between $\frac{s}{\delta s}\sqrt{N/2}$ and $\frac{s}{\delta s}\sqrt{2/N}$ due to multiplex noise propagation for the case of a monochromatic line and a continuum respectively (Englert et al., 2006). For the SHOW instrument, we observe absorption features within a micro-window that is isolated with a 2 nm bandwidth filter. In this case, the multiplex noise associate with a random relative pixel response in the spectral measurements is closer to the continuum case.





Although the relative pixel response behaves quasi-randomly in the interferogram domain, the relative response is a fixed pattern and the observed variability is fixed across the interferogram row for a fixed intensity level. Therefore, the observed noise pattern is also fixed in spectral space for the same signal level. This fact is used during the Level 0 to Level 1 data processing in order to characterize this error. It is shown that a secondary correction is feasible for the SHOW ER-2 airplane measurements that effectively minimizes the impact of this variability.

Assuming a negligible inter-pixel variability, the total uncertainty in the recorded interferogram signal (measured in DN) is approximated by Eq.4 where g is the gain factor (electrons/DN) used to convert the measured DN to electrons.

$$\sigma_j = \sqrt{\left(\frac{I_j + D}{g} + R^2\right)} \tag{4}$$

In the ideal case, the dark signal and readout noise are small relative to the photon noise. We cool the camera to reduce the dark signal and calibrate the remaining signal by subtracting a dark frame obtained by averaging hundreds of measurements. The readout noise and noise associated with the relative pixel response are characterized in the laboratory and are shown to be negligible compared to the Poisson noise. The signal to noise ratio (SNR) in the spectral samples is estimated using Eq.5 (Brault, 1983).

$$SNR_\sigma = \sqrt{\frac{1}{N}} \frac{B(\sigma)}{\overline{B_e}} SNR_x \tag{5}$$

In this equation, N is the number of samples in the interferogram, $SNR_x$ is the signal to noise ratio in the interferogram samples, $B(\sigma)$ is the spectral density at wavenumber, $\sigma$ and $B_e = \frac{1}{2}[B(\sigma) + B(-\sigma)]$.

## 7    Level 0 to Level 1 processing

The SHOW Level 0 data consist of the raw SHS interferograms and noise estimates, as well as, the airplane attitude information (time, geographical location, speed, pitch, yaw etc). These raw data are processed in several stages as shown diagrammatically in Figure 3. In the first stage, raw interferograms are corrected for dark signal and bad pixels (of which, there are very few) are removed by performing a nearest neighbor interpolation in the vertical dimension. Then, the flat field correction and non-uniformity correction are applied. Finally, the DC bias is removed from the interferogram signal in order to obtain the final Level 1A interferograms. In the second stage, corrected spectra are extracted from these interferograms. This consists of two primary steps. The first step is the application of Hanning apodization. The second step is the application of an FFT to each interferogram row to obtain the complex spectrum corresponding to each line of sight in the image. The amplitude of the complex spectrum is taken at each row to form a vertically resolved image of the water vapour absorption spectra.

As noted earlier, the airplane pitch varies as a function of time and this variation results in each row of the detector observing a range of tangent altitudes at the limb. As part of the Level 0 to Level 1 processing, the airplane attitude information is used to correct the interferograms for the "jitter" associated with the variation in the airplane pitch to generate a vector that contains





corrected lines of sight (LOS) corresponding to each row in each image. Therefore, the Level 1 data consists of the calibrated interferograms and spectra, as well as, the geometrical information required to map each interferogram row in the image to a particular tangent altitude at the limb.

5      The pitch variation of the airplane is also used to obtain a secondary correction that can be applied to minimize the impact of the systematic variations in the spectra associated with the residual relative pixel response in the interferogram signals. In practice, over a short period of time, it can be assumed that the radiance level at a particular altitude remains constant. Therefore, as the airplane pitches up and down, the signal level crosses different rows and this variability can be tracked in the spectral images. This allows this detector related systematic variability to be characterized and corrected in the measurements. This correction, discussed in detail in section 11.2, produces Level 1C corrected spectra.

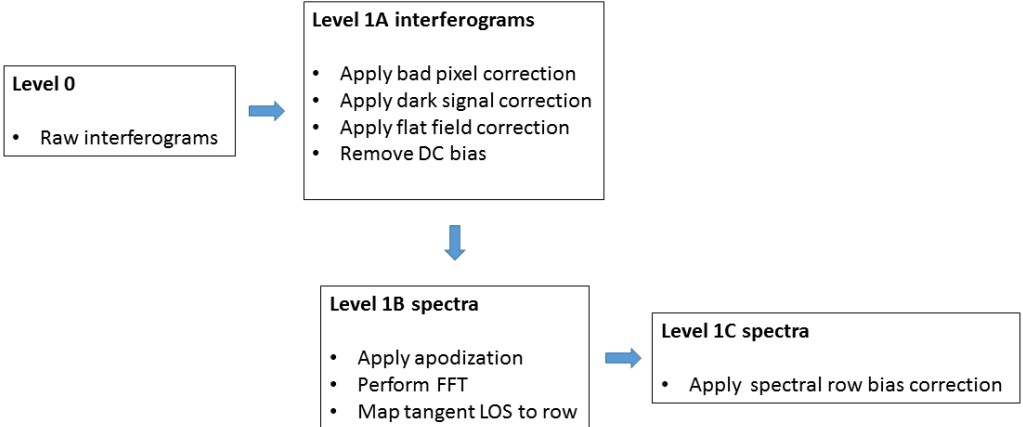

**Figure 2 SHOW Level 0 to Level 1 processing chain**

## 8      Retrieval approach

The SHOW retrieval algorithm utilized to process the flight data builds on the SHOW sensitivity study presented in a previous publication (Langille et al., 2018). The approach utilizes the non-linear optimal estimation formalism described in (Rodgers, 2000)

15      to extract water vapour profiles from the vertically resolved spectral measurements. The retrieval performs an iterative step given by Eq.6 in order to find the next best estimate of the state vector, x . In this case, x is the retrieved water vapour profile as the logarithm of number density.

$$x_{i+1} = x_i + \left(K^T S_y^{-1} K + R + \lambda I\right)^{-1} \{K^T S_y^{-1}[y - F(x_i)] - R(x_i - x_a)]\} \qquad (6)$$

In this equation, y is the measurement, i is the iteration number, $x_a$ is the apriori state vector, F is the forward model, K is the Jacobian matrix, $S_y$ is the measurement covariance matrix, $\lambda$ is a Levenberg-Marquart damping term, and R is a second order regularization matrix that is used to constrain the vertical resolution of the retrieval. The measurement vector, y, contains the logarithm of vertically resolved spectra that are normalized by their spectral mean. The forward model, F, is composed of the

SASKTRAN radiative transfer model (Bourassa et al, 2007; Zawada et al., 2015; Zawada et al., 2017) and the SHOW instrument





model (Langille et al., 2018). SASKTRAN is used to produce high resolution, forward modelled radiances and the SHOW instrument model is used to simulate SHOW measurements $F(x_i)$ from these modelled radiances.

The retrieval of the water vapour profile requires a forward model that accurately predicts the instrument behavior. For a typical FTS instrument, this is achieved by convolving the instrument line shape (ILS) with a high-resolution forward modelled spectra.

Generally, the ILS is obtained by taking measurements with a quasi-monochromatic calibration source. For the SHOW instrument, the convolution with the ILS at each row in the instrument produces a spectrum that does not capture the systematic variations associated with the aliasing effect discussed in Section 5. In the current work, we have performed a detailed instrument characterization to obtain appropriate calibrations that are utilized during Level 0 to Level 1 processing and we optimized an instrument model to accurately predict the systematic variations in the observed spectra. In Section 12, we utilize this model in

the retrieval to extract water vapour profiles from measurements that were taken during a SHOW ER-2 engineering flight on July 18 [th], 2017.

## 9      Laboratory Characterization

### 9.1      Overview

The SHOW instrument consists of several primary components. This includes, the field widened SHS, the optical system

(anamorphic fore optics and exit imaging optics), and the FPA. The configuration of these components introduces variations in the interferogram signals that must be characterized and calibrated in order to obtain the highest quality measurement. As noted in Section 5 and Section 6, treating each pixel in each interferogram row as a separate detector requires knowledge of the behavior of each pixel in the array. Converting the raw Level 0 interferograms to corrected Level 1 spectra requires detailed knowledge of the SHS configuration and the imaging quality of the instrument. In addition, knowledge of the vertical field of view is essential

to mapping the rows of the detector to the tangent altitudes at the limb. We also utilize the knowledge of the instrument configuration to develop the optimized instrument model that is applied to predict the systematic effects associated with aliasing.

| Detector parameter | Value |
|---|---|
| Bad pixels | 46 |
| Dark signal (1800 ms exposure) | 161 DN* |
| Dark noise (single frame) | 2-5 DN * |
| Dark noise (calibration frame) | < 0.5 DN |
| Bias | 1974 DN * |
| Gain | 45.7 e / DN * |
| Readout Noise | 165.7 e  (3.62 DN) |
| Non-linearity | < 1% |
| PRNU | < 1.2 % |

**Table 2 SHOW OWL 640 detector parameters characterized in the laboratory. Values with a * are representative of the average behavior**
**across the array**



## 9.2 Detector characterization

The FPA utilized in the SHOW-ER2 instrument is a commercial off the shelf (COTS) OWL 640 InGaAs detector manufactured by Raptor Photonics. The FPA has 640 x 512 pixels covering an area of 0.74 cm$^2$, 14 bit ADC, with a > 80 % quantum efficiency in the SHOW passband.  The temperature of the FPA is controlled to 0 degrees Celsius using a built in thermoelectric cooler in

order to reduce the dark signal. As noted earlier, bad/hot pixels, Poisson noise and the variation in the relative pixel response in the interferogram samples results in noise that propagates into the spectra.  These parameters have been characterized in the laboratory and are summarized in Table 2.

A total of 46 bad/defective pixels were identified within the imaged field of view (FOV) of the SHOW FPA. A handful of pixels around the edge of the FPA also exhibit odd behavior and have been identified as bad pixels.  During data processing, a bad pixel

map is utilized to identify these pixels and we interpolate over these pixels using nearest neighbor interpolation. To correct for dark signal we obtain average dark calibration frames for a range of exposure times by averaging several hundred dark measurements at each setting. The observed dark signal was found to vary non-linearly in the SHOW operating range using less than 2-second exposures. For the 0.5 Hz acquisition setting, the mean bias level of the detector was found to be approximately 1974 DN and the average dark signal across the frame using the 1800 ms exposure setting was found to be ~161 DN. The noise at

a single pixel in the average dark frames was found to be < 0.5 DN.  An average dark frame is subtracted from each flight measurement obtained with the same exposure time.

Propagation of error in the signal measurement using Eq.4 requires knowledge of the gain, readout error and the magnitude of the relative pixel response. The approximate gain (electron/DN) and readout error (R), calculated from an experimentally determined photon transfer curve, are 45.7 electrons/ DN and 165.7 electrons respectively. These values match closely with the values specified

by the manufacturer of 39.67 electrons/DN and 150 electrons. Therefore, the readout noise is estimated to be 3.62 DN (~ 4 DN). The detector was also found to exhibit a slight non-linear response to incident radiation. This non-linearity is on the order of 1 % across the full dynamic range of the detector. In practice, it was found that the non-linearity changed slightly with the detector operating settings (PCB temperature, FPA temperature etc.).This made characterizing the effect difficult; therefore, the non-linearity is not corrected in the calibration or flight measurements.

The relative pixel response was characterized in the laboratory by performing a two-point non-uniformity correction using flat field measurements. It was found that the PRNU associated with an uncorrected frame results in a ~1.2 % relative variation in the interferogram samples. The presence of the 1% non-linearity makes performing a non-uniformity correction difficult, since the extent of the linearity was found to change with the level of incident radiation, as well as, exposure time.   In the current work, the effect of the relative pixel response is minimized by obtaining flat field measurements using intensity levels that closely match

intensity level of the observations.  Instead of performing the non-uniformity correction, we correct the flight measurements using the flat field correction that best corrects for the relative pixel response variations at the observed intensity.

## 9.3 Imaging system characterization

The 4 degree vertical field of view by 5.1 degree horizontal field of view of the limb observed by SHOW is collimated at the entrance aperture and is passed through the SHS spectrometer using the anamorphic entrance optics. The optics are arranged to

produce a vertically resolved image of the limb at the grating location, whereas, the horizontal scene is averaged at the grating location. At the output, the detector and exit optics are mounted on a common holder that is aligned with the monolithic SHS. The aft imaging optics are arranged to form an image of the SHS interferograms conjugate to a vertically resolved image of the limb at



the detector. This is achieved by imaging the grating at the detector location. Therefore, the magnification of the exit optics determines the scale factor relating the observed spatial frequency on the detector to the spatial scale given by Eq.1. This parameter is used to define the bin spacing between the interferogram samples and ultimately the frequency scale of the observed spectra. In addition, this configuration establishes a one- to-one mapping between interferogram row and tangent altitude at the limb. Accurate

knowledge of these parameters is required for Level 0 to Level 1 processing.

### 9.3.1    Exit optics magnification

An example raw image obtained by illuminating the entrance aperture with white light from an 8" LabSphere integrating sphere is shown in Figure 3 (a). This image corresponds to the full frame provided by the (512 pixel x 640 pixel) SHOW detector. The contrast has been adjusted in Figure 3 (b) to highlight the SHOW FOV defined by the image of the aperture at the grating (red)

and the edge of the grating illuminated by scattered light (blue). The magnification of the exit optics is calculated by measuring image size of the grating (blue) on the FPA and taking the ratio of the known grating height to the measured grating height. For the SHOW ER-2 configuration, the magnification was measured to be 0.22.

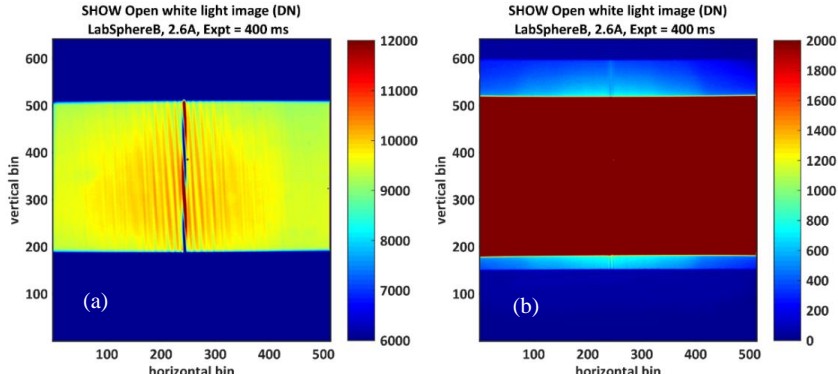

**Figure 3 An example dark corrected white light image (a) and the same image with a color scale that shows the SHOW field of view in**
**red and the edge of the grating in light blue (b)**

### 9.3.2    Field of view

Scattered sunlight from the limb is collimated at the input to the instrument, therefore, a particular off-axis angle illuminates a single row on the SHOW detector. As the off-axis angle is varied, the illuminated row also changes. Knowledge of this mapping is critical in order to determine accurate lines of sight that are used to obtain accurately georeferenced spectra. The SHOW field

of view only fills a portion of the detector frame – called the SHOW FOV from pixel 197 to 491 in the vertical and 9 to 502 in the horizontal. The outer edges of the red region in Figure 3 (b) have been removed from the FOV to ensure no edge effects are present. Later in the paper, only this FOV is shown, rather than the full frame that is shown in Figure 3.

This vertical field of view was characterized in the laboratory by directing a well-collimated beam into the instrument entrance at

different incident angles. For each incident angle, the position of the illuminated row was determined. A plot of the illuminated row number versus incident angle is shown in Figure 4. A LMS linear fit was performed to determine the slope of $m = -79.968\ pixels/deg$. The $R^2$ fitting coefficient has a value of 0.9996. From these measurements, the maximum angular vertical resolution of the measurement is ~0.0126 degrees. In practice, the FWHM of the imaged row was found to be approximately 1.4





pixels. Having 0.0126 degrees/pixel, this gives an instantaneous FOV for each sample of 0.0176 degrees. If we take the radius of the Earth to be 6371 km and assume the airplane to have zero pitch and an altitude of 21.33 km, this angular resolution converts to a tangent point vertical resolution of approximately 151 m at 2.5 km and 20 m at 21 km.

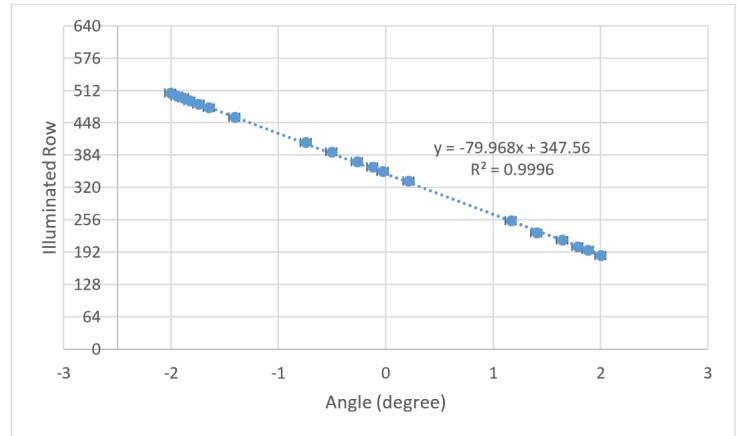

**Figure 4 SHOW measured vertical FOV**

### 9.4    Flat field calibration

The flat field calibration is performed using the approach developed by (Englert et al., 2006) and takes into account the flat field variations from the optics, as well as, slight differences between the two arms of the interferometer.  First, the entrance aperture of the instrument is uniformly illuminated with white light. Then, separate images are obtained with each arm of the interferometer

(arm A and arm B) blocked using an opaque material. Hundreds of measurements are obtained, dark corrected and then averaged to minimize the Poisson noise. Taking these images to be $I_A$ and $I_B$ respectively, the flat field term associated with optical effects is given by Eq.7.

$$FF1 = I_A + I_B \tag{7}$$

The correction term associated with relative differences between the two arms of the interferometer is given by Eq. 8.

$$FF2 = \frac{2\sqrt{(I_A \cdot I_B)}}{(I_A + I_B)} \tag{8}$$

Corrected interferogram images are obtained by dividing the dark corrected SHOW interferogram images by Eq.7, subtracting off the mean DC signal at each row and then dividing by the image generated using Eq.8.   If the FF2 term is close to 1, it can be neglected. In this case, the interferograms are corrected by dividing the interferograms by $\frac{FF1}{FF1}$ and then subtracting off the mean DC signal level at each row in the image.



Example FF1 and FF2 correction images that were obtained in the laboratory using an 8" aperture LabSphere integrating sphere are shown in Figure 5 (a) and Figure 5 (b) respectively. Optical variations of ± 5 % are observed in the FF1 image and the presence of dust and vignetting in the optical system is also apparent. The FF2 image is close to 1 and the variations are on the order of 0.1 %. Therefore, the SHOW SHS can be assumed to be well-balanced and the dominant correction term is due to optical flat field variations. This demonstrates the high quality of the optical system that was designed and manufactured by ABB Inc. and York University (Center for Research in Earth and Space Science), as well as, the interferometer that was designed and fabricated by Light Machinery.

For the SHOW data processing, we take the FF2 term to be equal to 1 and just apply the FF1 term which corrects for the optical variations. This also correct for the inter-pixel variations associated with the relative pixel response since the flat field images are obtained without performing a non-uniformity correction. However, the measured relative pixel response depends on the intensity of the incident radiation. Therefore, we obtain flat field correction images within the range of the intensity levels that are expected during flight. The flat field correction terms shown in Figure 5 were obtained with roughly quarter of the full well and an exposure time of 250 ms. The flight measurements are corrected using the calibration frames that are obtained at the closest matching intensity level.

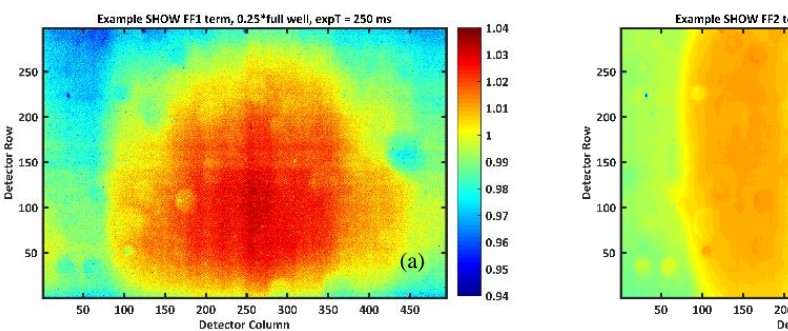
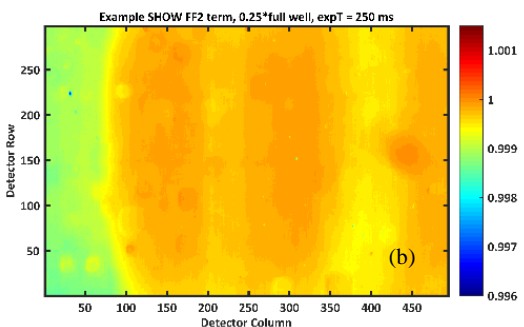

**Figure 5 Example SHOW flat field correction images obtained using a 250 ms exposure time to obtain a quarter full well signal. The FF1 term is shown in (a) and the FF2 term is shown in (b).**

### 9.5     SHS characterization

The primary parameters that are used to characterize the configuration of the SHOW SHS system are the Littrow wavelength and the spectral resolution. The SHS is designed to have an unapodized spectral resolution of: $\delta\lambda_{resolution} \simeq 0.02$ nm and a Littrow wavelength of 1363 nm (in air) at 20 degrees Celsius (Langille et al., 2017). This spectral resolution was calculated assuming that the full grating width is utilized. For the current version of the SHOW instrument, the imaged portion of the grating cross-section is slightly smaller - roughly 3.37 centimeters (494 pixels x 15um/pixel / 0.22). This gives a maximum theoretical resolution of ~ 0.03 nm where the maximum resolving power of the instrument is given by (Harlander, 1991): $R = 4W_g\sigma\sin(\theta_L) = 4\left(\frac{3.37}{\cos(28.5)}\right)(7336.8\ cm^{-1})\sin(28.5) = 53698$. In addition, the SHOW interferograms are apodized using Hanning apodization which results in a factor of ~1.8 increase in the spectral resolution (Davis et al., 2001). It is also known that the Littrow wavelength drifts in temperature by roughly 0.06 nm/ deg Celsius, and the flight instrument is operated at 23 degrees Celsius. Therefore, it was anticipated that the Littrow wavelength is higher than the design value by close to 0.18 nm.

Characterization of the SHS configuration was performed in the laboratory by uniformly illuminating the entrance aperture of SHOW with light from a Krypton calibration lamp. The Krypton lamp contains a well-isolated line at 1363.422 nm (in air) that

lies within the SHOW passband. Measurements with the Krypton lamp were used to measure the Littrow wavelength at the operating temperature and determine the spectral resolution of the instrument. Measurements of a white light source were utilized to characterize the full instrument. The instrument was purged with dry nitrogen for 24 hours prior to the characterization work in order to ensure a stable operating environment and an SHS temperature of 23 degrees Celsius.

An example image of the Krypton fringes obtained with SHOW is shown in Figure 6 (a). The rotation of the fringes is due to the presence of a slight tilt in the y-direction between the two gratings discussed in Section 2. The counter-clockwise rotation indicates that the Krypton line is located to the long wavelength side of the Littrow wavelength. The Littrow wavelength was determined by performing a non-linear least-mean squares cosine fitting to an interferogram row to determine the spatial frequency of the observed fringes. The spatial frequency of fringes measured on the detector surface is given by Eq.1, therefore the Littrow

wavelength is calculated from the observed spatial frequency and knowledge of the target wavelength. Using this approach, the spatial frequency of the observed fringes was found to be 1.98 fringes/cm and the Littrow wavelength was determined to be $\lambda_L =$ 1363.62 nm (in vacuum – calculated using the Ciddor equation) and 1363.25 nm (in air). This is close to the expected increase in Littrow based on the design of the instrument.

The Krypton spectrum, obtained by taking an FFT of an interferogram row close to the center of the image is shown in Figure 6

(b) in blue. The theoretical spectrum is obtained by taking the FFT of a simulated interferogram that is modelled by assuming a perfectly monochromatic line at 1363.422 nm is shown in red. Hanning apodization was applied to both interferograms prior to taking the FFT. The spectral resolution was estimated by determining the FWHM from a Gaussian fitting to the spectral line. The FWHM of the measured spectrum is 0.0516 nm and the FWHM of the theoretical line is 0.0451 nm. The two values differ by roughly 0.0065 nm and the difference is likely due to slight distortions in the spectrum that are introduced by the imaging optics.

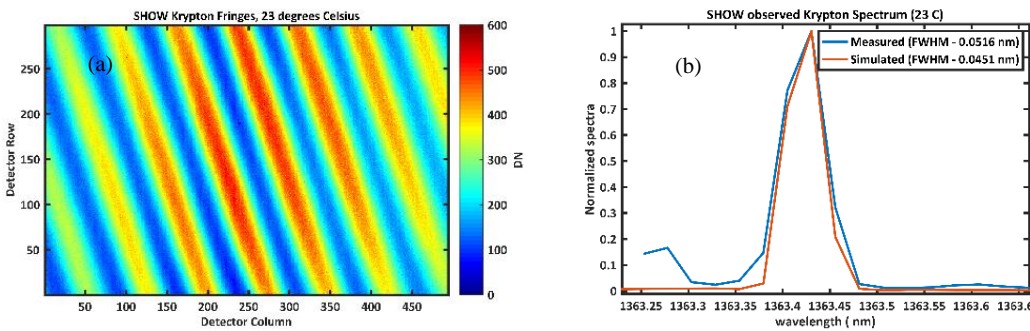

**Figure 6 Krypton fringes observed with the SHOW SHS instrument at 23 degrees Celsius (a) and the spectrum obtained by taking the FFT of the interferogram at row 150 (b). The measured spectra is shown in blue and the simulated spectrum is shown in red.**

### 9.6    Performance

The performance of the SHOW flight instrument was characterized in the laboratory by uniformly illuminating the instrument with

white light from an 8 " aperture LabSphere integrating sphere. One hundred and fifty interferogram images were obtained and then processed using the Level 0 to Level 1 processing chain to obtain calibrated interferograms and corrected spectra. An example calibrated interferogram image is shown in Figure 7 (a) and an example raw and corrected interferogram row is shown in Figure 7 (b). The corresponding spectral image formed by taking the FFT of each row of the interferogram image is shown in Figure 7 (c). All rows of the spectral image are plotted in Figure 7 (d). The SHOW filter is a narrow band filter (see Langille et al., 2018) with

a 2 nm bandwidth and a peak transmission of 0.77 that is centered at 1364.52 nm at 23 degrees Celsius. The shape of the filter





passband is clear in Figures 7(c-d) and the sharp cut off in the spectra on the left hand side corresponds to the location of the Littrow wavelength at 1363.62 nm (in vacuum).

The goal here is to examine the quality of the corrected interferograms and spectra in order to estimate the remaining variability in the images that is not associated with photon noise. It is difficult to characterize the noise in each individual interferogram and

5    spectral row since samples within each row contains variations associated with the source. However, the vertical dimension of the interferogram and spectral image is expected to be smooth since the illumination within the field of view is uniform and the optical configuration is anamorphic. Therefore, we examine the inter-sample variability in each vertical column in order to characterize the noise.

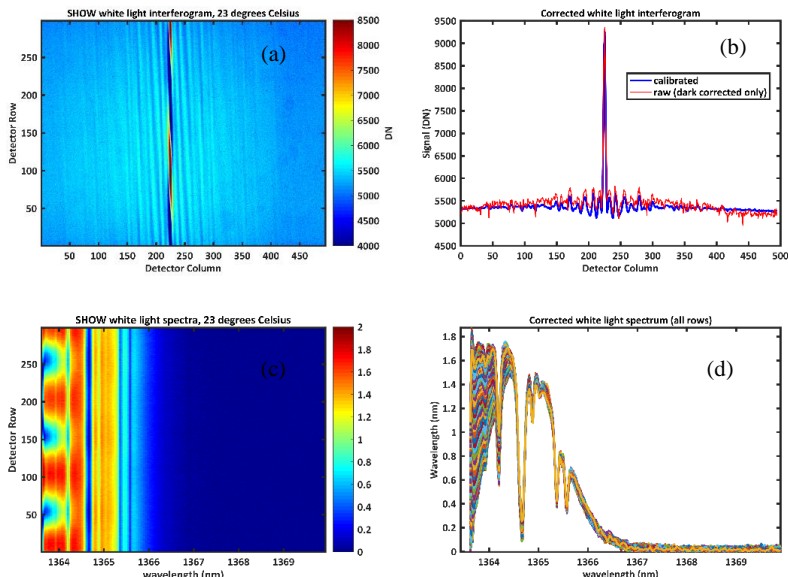

**Figure 7 Example corrected white light interferogram (a) and a row cut taken at row 201 (b). The dark/bad pixel corrected signal is shown in red and the dark/bad pixel and flat field corrected signal is shown in blue. Example spectral image obtained by taking an FFT of each row in the interferogram image (c) and all rows of the spectral image plotted (d)**

Ideally, we want to operate the detector with the minimum amount of noise. Taking the variance on any particular interferogram sample to be given by Eq.4, we estimate the noise floor by averaging the images to reduce the photon noise. For an average signal level of ~ 5300 DN and averaging 192 images, we find the noise floor for the samples to be roughly σ = 10.7 DN. Taking into account photon noise, dark noise, and readout noise, we estimate the observed noise associated with the relative pixel response

20    (using Eq.4) to be approximately 0.2 %. This is an order of magnitude smaller than the PRNU of an uncorrected image and demonstrates the quality of the combined flat field calibration and non-uniformity correction.

On the other hand, the observed variations in the spectral image is slightly more difficult to quantify. We expect that each row in the image will record the same spectrum; however, this is not what is observed. Instead, we observe a strong modulation of the spectra in the vertical dimension. This variation is systematic and is due to the presence of a small relative tilt between the two

25    gratings in the SHS combined with aliasing of spectral information from the left side of Littrow into the right hand side (see Section



5). This systematic variation is characterized in the next section by optimizing an instrument model that predicts the effect. Here we focus on the propagation of error from the interferogram samples to the spectra.

Since each row of the interferogram should produce the same spectrum, the presence of the inter-sample variability in the spectra is determined at each wavelength (column) by subtracting off a high order polynomial from each column cut and then determining

the standard deviation of the subsequent distribution. Following this approach, we determine the inter-sample variability in the vertical dimension to be roughly 3 %. This noise is primarily associated with relative pixel response in the interferogram samples; therefore, the spatial variation is fixed within any particular image. In Section 11.3, we show that the pitch of the airplane and the subsequent changes in intensity level across the rows, can be used to further correct this variation.

**9.7      Model optimization**

The aliasing effect that is shown Figure 7 (c-d) is extremely problematic for the retrieval of the vertical distribution of water vapour. The observed variation modulates the true spectrum and the amplitude of the effect depends on the relative position of the filter center peak and the Littrow wavelength, as well as, the amount of absorption due to water vapour. In order to implement the retrieval approach described in Section 8, a forward model is constructed that captures these perturbations. The model takes a

simulated high-resolution spectrum $B(\sigma)$ that is simulated using SASKTRAN and constructs the interferogram image given by Eq.9.

$$I(x,y) = \int_0^\infty \tau(\sigma)B(\sigma)[1 + +\cos[2\pi(4(\sigma - \sigma_0)(x - x_{shift})\tan(\theta_L) + \alpha\sigma(y - y_{shift}))]]d\sigma \qquad (9)$$

This equation follows directly from Eq. 3, where we have assumed the interferogram has been dark and flat field corrected and we have included a linear phase variation in the y-direction associated with the grating cross tilt, as well as, a relative shift to both the x and y dimensions. These relative shifts account for a lateral detector misalignment or a shift in the relative axis of rotation between the two gratings. Each row of the modelled and measured spectral images is normalized by the mean of the full spectrum at each row to ensure the spectra are on the same scale.

As a first step prior to performing the optimization, the Littrow wavelength ($\sigma_o$) was determined by aligning the observed water absorption features in the white light spectra with known features in the high-resolution simulated spectrum. In order to model the correct amount of absorption, we need to estimate the water vapour abundance, as well as the path length. For this calculation, we assumed standard temperature and pressure and a relative humidity of 50 %. The path length was then manually adjusted in the model until the depth of the absorption features were closely matched.

The instrument model was then optimized by iteratively adjusting the center peak of the filter, the value of the grating rotation $\alpha$, and the relative shifts in both the $x$ and $y$ positions on the grating. After each iteration, the spectral agreement between the model and the SHOW white light measurements was checked by calculating the average absolute difference in the spectra signal, as well as, the standard deviation of the difference in the systematic variations introduced by the aliasing effect. The parameters used on the iteration which produces the best agreement between the model and SHOW is then taken to be the best representation of the

true configuration of the instrument for this measurement.





Figure 8 (a) shows the final agreement between the instrument model and the SHOW white light laboratory measurements as a result of this process. The normalized interferograms are shown in the top panel and the normalized spectra are shown in the bottom panel. Here, the image has been rotated so the lower rows correspond to the lower altitudes when compared to the flight measurements. For this set of measurements, the Littrow wavelength was found to be 1363.62 nm (in vacuum), the filter center

was found to be 1364.52 nm, the grating tilt was found to be -5.108 x 10$^{-5}$ radians, and the x, y offset was found to be ( 15.77 pixels, -15.52 pixels) respectively. Two rows corresponding to the best and worst agreement are shown in Figure 8 (b) and Figure 8 (c). The blue line in both of these figures shows the forward modelled high-resolution water vapour spectrum. There is reasonable agreement between the model and measurement and the model appears to capture the shape of the systematic variations associated with aliasing. Unfortunately, it is difficult to quantify the differences between the spectra since the amount of water vapour in the

laboratory (and our lack of knowledge of it) has a strong influence on the amplitude of the variations; especially close to the absorption features. We will come back to the optimization in the case of flight data where we have more information regarding the water vapour abundance via the in-situ radiosonde measurements.

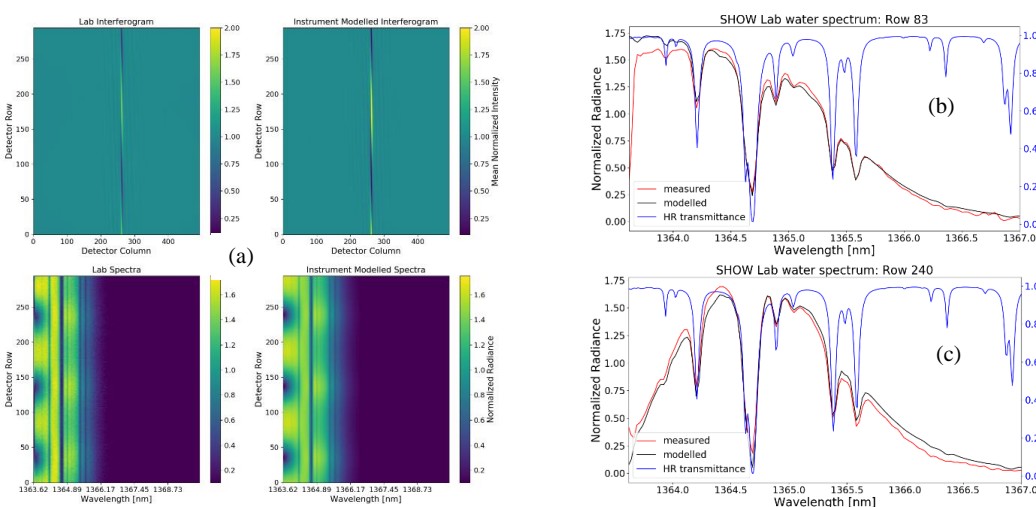

**Figure 8 Instrument model optimization using laboratory data. Comparison between the modelled and measured interferograms and spectral images (a). Example spectral row cuts at row 83 (b) and at row 240 in (c). The measured spectrum is shown in red and the modelled spectrum is shown in black. The normalized high-resolution spectrum that was utilized in the instrument model is shown in blue.**

## 10      SHOW demonstration flight on July 18th, 2017

On July 18th, SHOW flew an engineering flight on the ER-2 airplane in order to test several aspects related to the performance of the instrument. The flight track of the ER-2 airplane for the entire engineering flight is shown in Figure 9 (a) overlaid on top of a map showing the geographic location of the aircraft above California. The airplane attitude information during the flight is shown in Figure 9 (b) where the recorded airplane pitch, roll, heading, height, latitude and longitude are shown as a function of time. Portions of stable flight correspond to regions where the pitch and roll are relatively close to zero; although small variations in the

pitch are always recorded – even for the most stable portions of the flight. In practice, this airplane attitude information is used to map the interferogram rows to tangent altitude at the limb for each frame obtained with the SHOW instrument.





Prior to the flight, the instrument was mounted in the wing pod (detached from the airplane) and was purged with nitrogen for over

24 hours in order to remove moisture and to achieve a stable thermal environment (23 degrees Celsius) within the instrument optics

box. About an hour before the flight, the instrument was turned off and wheeled out to the tarmac and the wing pod was mounted

to the airplane. The temperature on the tarmac was roughly 25 degrees Celsius during this time and the SHOW instrument and

5    thermal control was powered on just before takeoff. The recorded instrument temperatures inside the optics box (SHS and

enclosure) and inside the wing pod (periscope and wind pod window) during the complete engineering flight are shown in Figure

10(a) and Figure 10 (b) respectively.

(a)

(b)

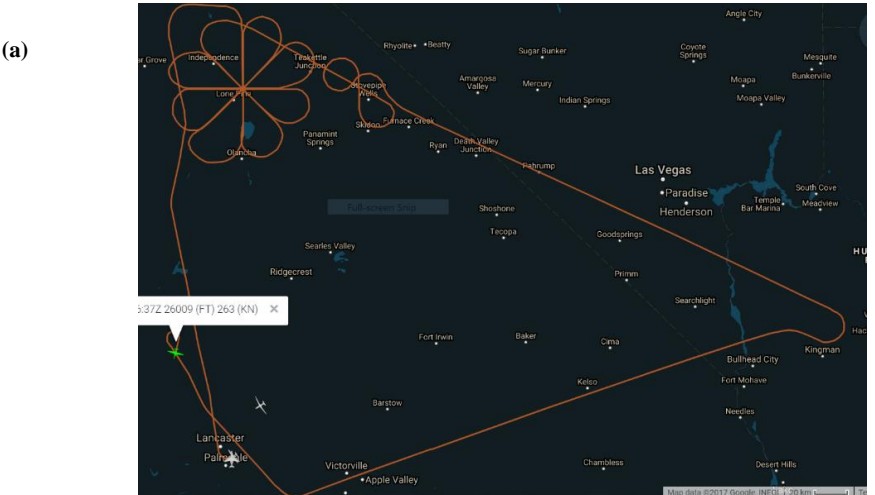

**Figure 9 ER-2 flight track (a) and the corresponding flight attitude information (b) for the engineering flight performed on July 18, 2017**




We can see that the temperature inside the wing pod at the beginning of the flight is relatively warm, corresponding to the warm morning temperatures on the tarmac at takeoff. Once the airplane reached its cruising altitude, the temperature dropped steadily and reached close to – 40 degrees Celsius at the wing pod window and close to – 20 degrees Celsius at the SHOW periscope. During this same period, the temperature inside the optics-box initially increased to 24.5 degrees Celsius and then steadily

decreased towards 22 degrees Celsius.

The thermal control is designed so that the heaters on the top and bottom of the optics-box keep the top and bottom center area of the SHS close to 23°C; however, we observed that the SHS temperature settles closer to 24.3 degrees Celsius once the optics-box temperature begins to stabilize. Since the thermal time constant of the SHS is quiet long, we suspect that the SHS was still stabilizing after the large rapid increase in the temperature inside the optics box at the beginning of the flight. We believe that the

SHS temperature would have begun to decrease towards 23 degrees Celsius over a longer period. In any case, the temperature of the SHS varies by roughly 1.5 degrees Celsius during the flight and the temperature of the SHS remained relatively stable for the last hour of the flight. In practice, the Littrow wavelength is deduced by lining up the known water vapour absorption features with the high resolution forward modelled spectrum during data processing.

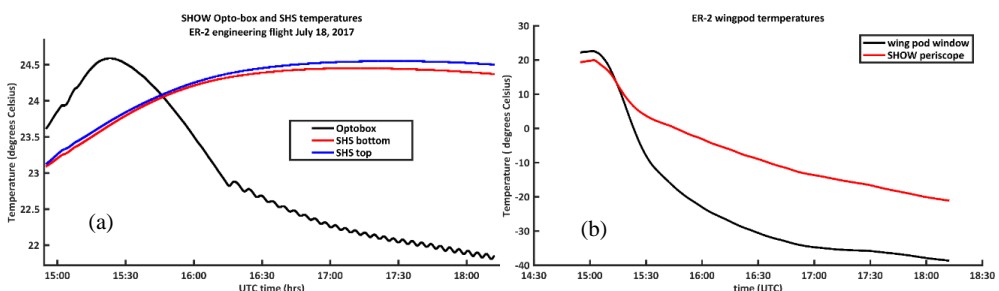

**Figure 10 SHOW SHS and opto-box temperatures during the course of the flight (a) and the ER-2 wing pod temperatures recorded at the wind pod window (black) and the SHOW periscope (red)**

Following from Figure 9 (a-b), the ER-2 airplane took off from AFRC at roughly 15:02 UTC and flew northward where the pilot performed a set of flight patterns designed to examine the sensitivity of the instrument to the observed scattering angle. Afterwards, the airplane flew towards the southeast and then turned to fly in the direction towards Table Mountain. During this time, SHOW was configured to continuously record images with integration times of 1800 ms at 0.5 Hz sampling. At 17:59 UTC, a radiosonde (model Vaisala RS41) was launched from the JPL facility located close to Table Mountain to measure relative humidity,

temperature and pressure as a function of altitude. The in-situ water vapour abundance was deduced from these measurements using the Hyland and Wexler formulation (Hyland and Wexler, 1983). Roughly 12 minutes of coincident measurements (over 300 samples) of the same approximate column of air were obtained. The in-situ water vapour abundance that was measured by the radiosonde is shown in Figure 11. The abundance increases from 5 ppm at 20 km to roughly 35 ppm at 13 km. In general, the uncertainty on the radiosonde measurement is less than ± 1ppm.

In the remainder of this paper, we focus on the set of measurements performed between 17:59 UTC and 18:09 UTC (bracketed by the red vertical bars in Figure 9 (b)). During this period, the airplane remained stable with pitch variations of < ±0.5 degrees and



a mean sea level altitude of ~20.85 km. The temperature of the SHS was also stable during this time. According to the radiosonde measurements, the water vapour abundance increases rapidly below 13 km (not shown in the Figure), therefore, the optical thickness becomes exceeding large making it difficult to observe the tangent altitudes below that level. Therefore, the useable field of view within this range is from 13 km to 21 km, corresponding to roughly 42 % of the SHOW limb image.

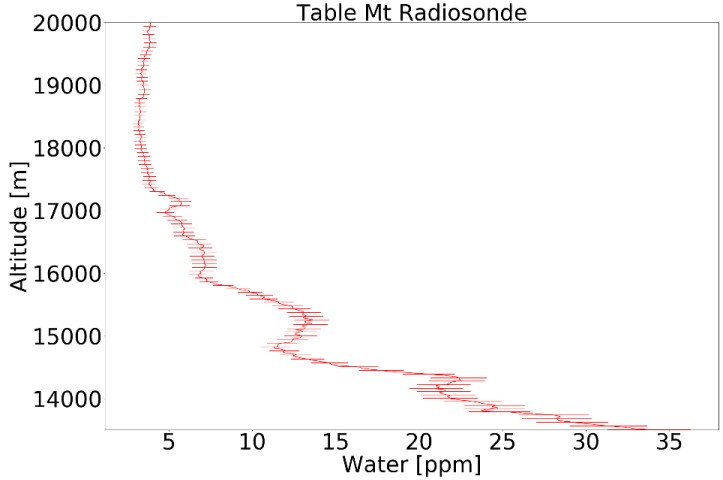

**Figure 11 Measured in-situ water vapour above Table Mountain using the JPL Vaisala RS41 radiosonde measurements.**

## 11    Flight data processing

### 11.1    Level 1A interferograms

Each image obtained by SHOW during the Table Mountain portion of the flight is corrected using the Level 0 to Level 1 processing
chain illustrated in Figure 2. An example of the correction of a raw interferogram is shown in Figure 12. The raw interferogram is shown in Figure 12 (a) and the fully corrected interferogram is shown in Figure 12 (b). In these figures, the SHOW image has been rotated so the bottom part of the image corresponds to lower tangent altitudes. Figure 12 (c) shows the application of the correction for the interferogram row – 150. Here the DC bias has been removed by subtracting off the mean of each interferogram row in order to isolate the modulated component of the interferogram. These interferograms represent the SHOW Level 1A data
product.

In this particular measurement, the presence of clouds is seen in both the raw, as well as, the fully corrected modulated component. There are several primary effects on the SHOW measurements associated with the presence of clouds. The most obvious is the observed horizontal variability in lower part of the non-modulated component shown in Figure 12 (b). Ideally, the anamorphic optics averages over the horizontal variability in the scene radiance and the flat field correction assumes a uniform illumination of
the input aperture. Here we observe that the variability associated with the cloud appears to be slightly different on either side of the interferogram center burst. This suggests that horizontal scene information results in a non-uniform illumination of the entrance aperture, resulting in a poor flat field correction up to row 100 (~13.5 km) in the FOV. Clouds also have a large optical depth, making it difficult to see the tangent point at the limb. Therefore, we do not expect to be able to perform retrievals on this particular measurement below 13.5 km.





In some cases, scattering from clouds saturates pixels at the interferogram center burst, resulting in corrupted pixels throughout the image. The presence of the saturation effect associated with clouds is detected by observing a small number of pixels off to the edge of the detector, far away from the SHOW FOV, as well as, the average signal across the FOV. This region also acts as a dark current and bias monitor during the flight. In practice, it was found that spikes in the number of saturated pixels correlate with

spikes in both the FOV average and the off-image average. For the measurements presented in this paper, images that have strong cloud effects are identified and removed from the analysis.

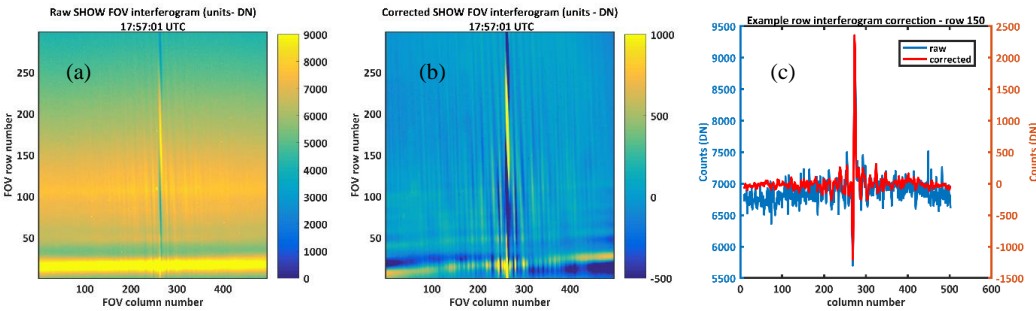

**Figure 12 Example conversion of raw to Level 1A interferograms from raw images (a) to corrected interferograms (b). An example row**
**cut at row 150 is shown in (c).**

The quality of the corrected interferogram has been characterized by examining the observed variability in column cuts of the image using the same approach that was used in Section 9.6. We understand that this approach is only approximate, since the variance of the measurement varies as a function of signal level. From Figure 12 (a), we can see that the raw interferogram signal

varies from ~4000 DN at high altitudes (higher row number) up to ~ 8000 DN at lower altitudes (lower row number). The dark and bias corrected interferogram varies from 2000 DN up to 6000 DN.

We estimate the measured variability in the image by subtracting a high order polynomial from each column in the interferogram to remove the atmospheric radiance profile and then calculating the standard deviation of the difference between the fit and the measured interferogram signal. The measured variability is roughly ± 23.48 DN. Assuming a mean signal level of 3700 DN, dark

noise of 0.5 DN and a readout noise of 3.62 DN (see Table 2) we estimate the noise associated with the relative pixel response to be roughly 0.6%. This value is approximately 3 times higher than the random relative pixel response that was characterized in the laboratory. The increase is likely due to the fact that the flat field correction is chosen to match the mean intensity level and will not completely correct for the relative pixel response at the different intensities.

### 11.2 Airplane-pitch correction

As noted earlier, the field of view of the instrument at the limb is mapped one-to-one to a particular row on the detector. Therefore, the same interferogram row observes a range of different tangent altitudes as the airplane pitches up and down during the flight. The retrieval algorithm described in Section 8 utilizes a forward model which requires accurate information regarding the viewing geometry of the instrument. In the worst case, large errors in the knowledge of the viewing geometry will result in a forward model that does not accurately simulate the radiative transfer problem for this volume of the atmosphere. In the case of smaller errors, the

result is a degradation of the vertical resolution of the measurement for any particular frame.





The ER-2 airplane pitch variability and its impact on the SHOW measurements during the Engineering flight on July 18, 2017 is demonstrated in Figure 13 (a) where we have plotted the signal measured by a single pixel as a function of time along with the ER-2 pitch variation. It is very clear that a strong correlation exists between the temporal variability of the interferogram signal and the pitch variation. In this particular case, the interferogram signal is anti-correlated with the pitch variation, corresponding to

the location of detector pixel [150, 205] (see Figure 12 (a)) in the SHOW FOV, where we have a lower atmospheric signal above and a higher atmospheric signal below the corresponding tangent altitude. It is also clear that the pitch variation is slow relative to the 0.5 Hz sampling cadence of the SHOW measurements.

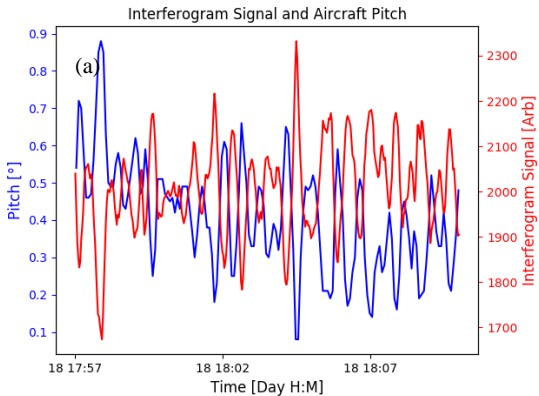

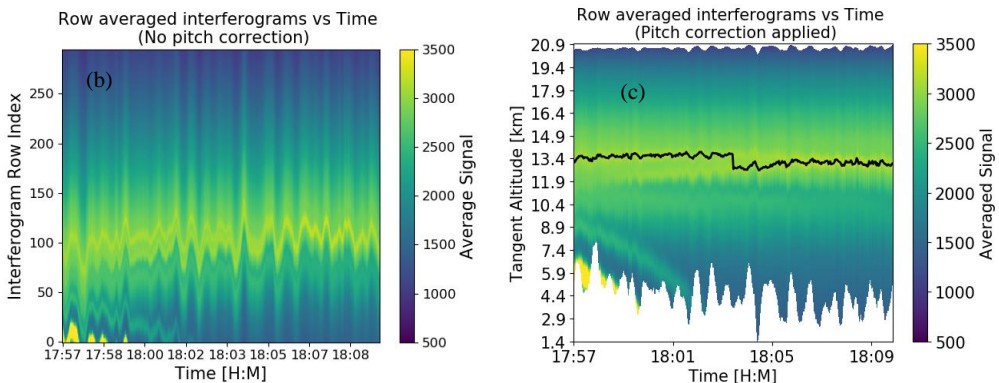

**Figure 13 Example of the correction for the airplane pitch variation. The interferogram signal at a single pixel is plotted with the pitch variation as a function of time is shown in (a). The average signal at each interferogram row plotted as a function of row number and stacked as a function of time is shown in (b). The average signal at each interferogram row corrected for pitch variation and plotted as a function of tangent altitude and stacked as a function of time is shown in (c) and the solid black line is described in the text.**

The pitch variation is further demonstrated in Figure 13 (b), where the mean signal level of each interferogram row is plotted as a function of altitude and then stacked as a function of time. For each row in each frame, the signal at each vertical bin corresponds to the DC component of the corresponding interferogram row. Therefore, the vertical variation of these samples in each frame provides a rough approximation of the shape of the integrated atmospheric radiance profile. In the ideal case, without pitch variation, we expect this profile to be relatively smooth, both temporally and spatially over this short 12-minute window. For the

SHOW ER-2 measurements, we see that the airplane pitch variation shows up as vertical bands that correspond to the detector rows scanning up and down the atmospheric radiance profile. We also observe the presence of several cloud features below row


100 with the dominant cloud feature sitting between row 80 and row 120 and persisting through the full observation period. The impact of the pitch variation is the most obvious at the location of the clouds since the gradient in the limb radiance is largest at the cloud feature.

Since the pitch variation is slow relative to the 0.5 Hz sampling cadence, we correct for the pitch variation by using the airplane

attitude information to map each interferogram row to the corresponding tangent altitude at the limb as shown in Figure 13 (c). This removes a significant portion of the variability; however, a residual temporal variation on the order of 1% of the mean signal level remains. The primary impact of the presence of this variability is the reduction in vertical resolution associated with the error in the pointing knowledge. This degradation is quantified by examining the vertical perturbations in the two dominant thin cloud features located at mean altitudes of 13.6 km and 13.1 km respectively. The black line shows the location of the peak signal

associated with these features. The standard deviation from the mean altitude level is found to be 124.45 m for the feature at 13.6 km and 175.03 m for the feature at 13.1 km. Assuming a mean ER-2 altitude of 20.85 km, these perturbations correspond to an angular uncertainty in the pointing of 0.023 degrees and 0.032 degrees for these two cases. Taking the maximum of these perturbations to be representative of the pointing error, we estimate the uncertainty in the pointing knowledge to be ~0.032 degrees. This is larger than the, 0.0176 degree, instantaneous tangent point angular resolution provided by the instrument. Therefore, we

find the ER-2 measurements obtained during the engineering flight to have a slightly degraded vertical resolution.

### 11.3 Level 1B spectra

SHOW Level 1B spectra are generated from the Level 1A interferograms by applying Hanning apodization to each of the interferogram rows, performing an FFT and calculating the amplitude spectrum. Each row of the spectrum is mapped to a tangent

altitude at the limb using the pitch correction described in the previous section. Therefore, a single record consists of the Level 1B spectrum and a mapping of spectral row to the tangent altitude at the limb. An example Level 1B spectrum obtained by applying this procedure to the interferogram image obtained at 17:59:05 UTC is shown in Figure 14 (a). The wavelength registration is set using knowledge of the instrument configuration and by manually adjusting Littrow wavelength to line up the absorption features with a high-resolution forward modelled spectrum. For this particular measurement, the Littrow wavelength was found to be

1363.76 nm. Over the course of the 12-minute Table Mountain portion of the flight, the Littrow wavelength was not found to vary significantly as a function of time.

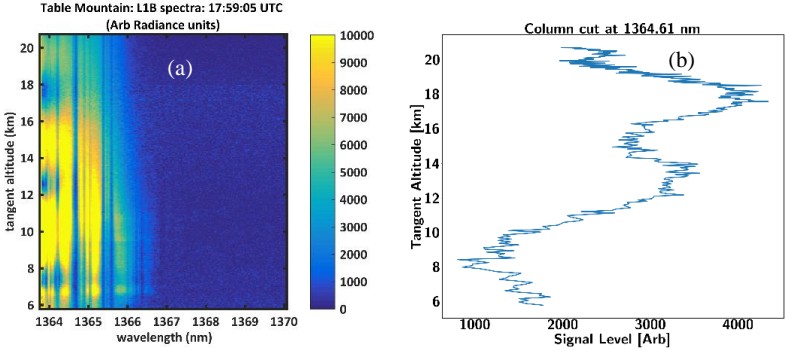

**Figure 14 Example Level 1B spectral image (a) and a column cut taken at a wavelength of 1364.61 nm (b)**





Figure 14 (b) shows a column cut at a wavelength of 1364.61 nm. Several sources of variability are apparent in the spectral samples. First, note the natural variability associated with the limb scattered radiance profile. The shape of this profile depends heavily on the atmospheric state (aerosol content, the water vapour abundance and clouds etc.). This particular wavelength sits in the wing of a strong absorption feature that is centered near 1364.66 nm. In this case, the radiance profile is modulated by the systematic

variations in the spectrum introduced due to aliasing effects. This effect was anticipated earlier in the paper and is characterized in the next section. Here we focus on characterizing the signal-to-noise ratio of the corrected spectral samples.

The two dominant sources of noise in the measurements are Poisson noise, as well as, a systematic inter-sample variability, both of which are introduced by the propagation of error into the spectrum from the interferogram samples (see Section 6). The noise in the interferogram samples is calculated using Eq.4 and the associated noise in the spectral domain is calculated using Eq.5.

However, as we shall now show, the impact of the relative pixel response on the spectral measurements can be minimized by utilizing the temporal series of samples obtained while the airplane pitches up and down to obtain a secondary calibration that can be applied to the spectral measurements. In this case, the noise on individual spectral samples is primarily due to Poisson statistics.

For this characterization, we assume that the shape of the true atmospheric profile is smooth and constant. This is a justified assumption given the stability of the observed integrated intensity profile shown in Figure 13 (c). Therefore, as the airplane pitches

up and down, multiple rows in the SHOW FOV observe the same atmospheric signal. For example, Figure 15 (a) shows the raw non-normalized L1B spectral signal level at a wavelength of 1364.81 nm for each detector row that observed the tangent altitude at 13.122 km (shown in black) within the 12 minute observation window. The black data points form clusters at each detector row that are approximately randomly distributed. This distribution includes Poisson noise, as well as, noise due to the temporal variability in the signal level associated with natural variability and errors in the pitch correction (see Section 11.2). The error bars

show only the Poisson distributed spectral noise.

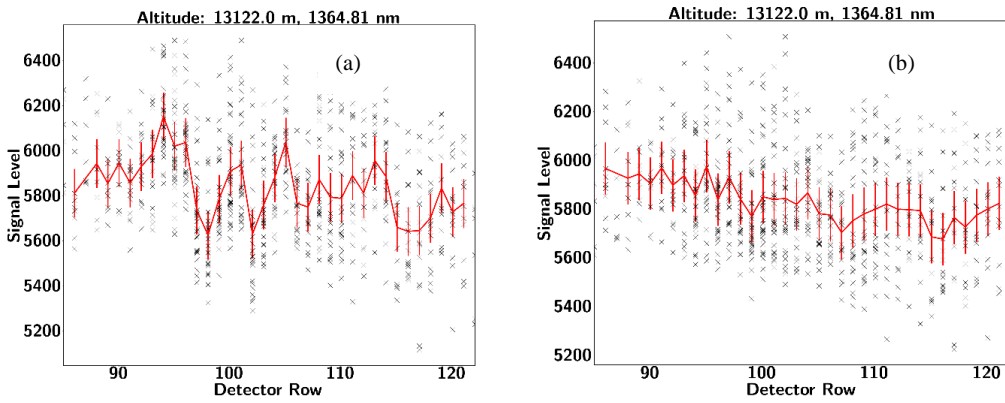

**Figure 15 Observed systematic inter-sample variability in the measured spectra for a single wavelength (1364.81 nm) and altitude (13.122 km) (a). The same information is plotted in (b) after the secondary correction is applied. The error bars denote Poisson noise and the**

**average SNR of the measurements for this set of samples is 53.83.**

Ideally, each row will register the same radiance; however, we observe that each cluster exhibits a slight offset for each detector row. These offsets were anticipated in Section 6 and are due to the propagation of error associated with the relative pixel response



across the interferogram rows. For these particular measurements, the systematic variability is of the same magnitude as the combined temporal variability and Poisson noise.

The observed systematic biases are isolated in several steps. First, each of the sample clusters is averaged to minimize the random noise (shown in red) and clusters that have fewer than ten samples are discarded from further analysis. We then normalize the averaged values by the mean signal level across all of the detector rows that observed this altitude. This data is then merged with the normalized averaged clusters from all the other tangent altitudes in order to isolate the inter-row biases for this particular wavelength. In order to remove the larger-scale variation associated with aliasing, we divide out a high order polynomial from the resulting data. This procedure is repeated for all of the measured wavelengths to obtain a correction for each spectral column in the image. Therefore, we form a secondary spectral image correction that can be applied to the Level 1B spectral images to minimize the systematic inter-sample variability.

The results of applying this correction to the L1 B spectral images is demonstrated in Figure 15 (b) where we plot the same data as shown in Figure 15 (a), this time with the correction applied to the spectral measurements. In this case, the amplitude of inter-row sample variability has been reduced by a factor of roughly 5 and this variability is now significantly smaller than the noise due to Poisson statistics. Assuming only Poisson noise, the average SNR of this set of samples is 53.83.

Each Level 1B spectral image obtained during the Table Mountain portion of the flight was corrected using this approach in order to obtain Level 1C spectra. An example of the impact of the secondary correction on an individual spectral row and an individual spectral column within a single image is shown in Figure 16 (a) and Figure 16 (b) respectively. While the correction has a minimal impact on the wavelength dimension, it has a significant impact on the vertical variability. This correction minimizes the impact of the systematic inter-sample variability; therefore, the noise on the corrected spectral samples is taken to be Poisson distributed and this noise level is included in the retrieval algorithm in order to compute the associated measurement error of the retrieved vertical water vapour distribution.

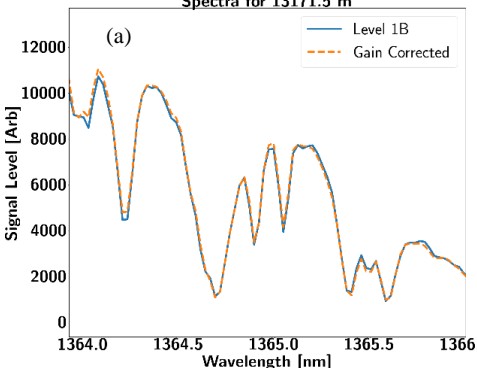
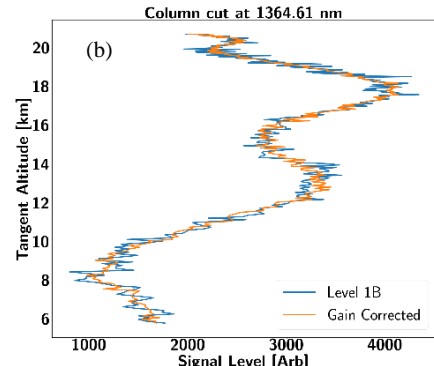

**Figure 16 Correction of the L1B spectra. Variability before and after the secondary correction is applied for an example row spectrum at a fixed altitude (a) and for a single wavelength column versus altitude (b)**





### 11.4 Instrument model optimization

The systematic perturbations to the radiance profile associated with aliasing are characterized by applying the optimization approach that was used in Section 9.7. However, for the Table Mountain flight measurements, we use in-situ water vapour measurements as the input to the forward model. Knowledge of the in-situ water vapour abundance provides a better representation

of the true atmospheric state and allows for a better quality optimization to be performed. Forward modelled interferograms and spectra are generated and compared to the flight measurements and the instrument model is iteratively adjusted to identify the instrument configuration that best predicts the measurements.

The results from this optimization are shown in Figure 17. The optimized configuration corresponds to a Littrow wavelength of 1363.76 nm (in vacuum), a grating tilt of -5.085 x 10$^{-5}$ radians, a filter shift - 0.048 nm, and x, y offsets of ( 17.27 pixels, -18.58

pixels) respectively. Two spectral rows are shown in Figure 17 (b) and Figure 17 (c) corresponding to the tangent altitudes of 9.878 km and 19.993 km respectively. There is general agreement between the forward modelled spectra and the measured spectra; however, the best agreement is observed from 1364.52 nm to 1365.67 nm. The presence of clouds below 13.5 km also makes the retrieval of water vapour below this altitude difficult. Therefore, the best region to target for a SHOW retrieval is from 1364.5 nm to 1365.67 nm and from 13 km up to 18 km. In this window, the observed relative variability between the measured and forward

modelled spectra is less than ± 5 %.

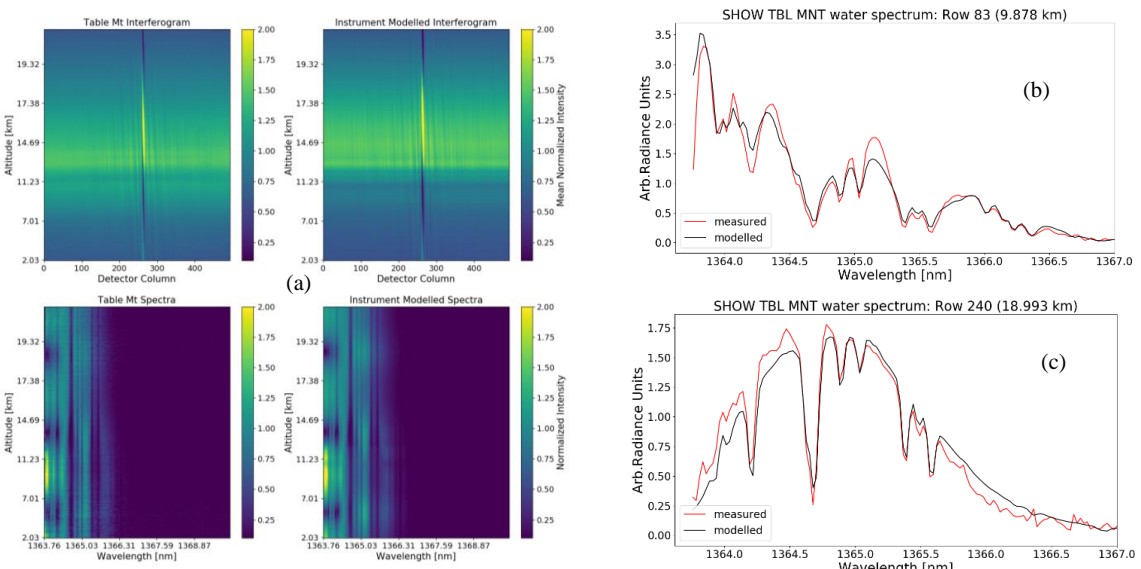

**Figure 17 Instrument model optimization using flight data. Comparison between the modelled and measured interferograms and spectral images (a). Example spectral row cuts at row 83 (b) and at row 240 in (c). The measured spectrum is shown in red and the modelled**
**spectrum is shown in black.**

### 12 Example water vapour retrieval

For the water vapour retrieval, we focus on a 10-image measurement window from 15:57:29 UTC to 15:57:49 UTC and average the frames to increase the SNR of the measurements. During this time, SHOW has good coincidence with the radiosonde measurements and the aircraft is pitched slightly upward above the cloud features that are present at the beginning of the Table





Mountain portion of the flight. The retrieval is performed using the approach discussed in Section 8. For the retrieval, we take the upper and lower altitude cutoff to be 18 km and 13.5 km respectively. The upper cutoff was chosen to be a few kilometers below the airplane altitude. Above this altitude, the region is optically thin and produces weak spectral signatures that reduce the accuracy of the retrieval. The lower cutoff was chosen to be above the region with clouds and above the region where the atmosphere is too

optically thick to observe the tangent point.

The SHOW measurements utilize the full spatial resolution that is provided by the vertical sampling within the imaged FOV. However, it was shown in Section 11.2, that the instantaneous angular vertical resolution is degraded to 0.032 degrees due to uncertainty in the pointing. Therefore, within the target retrieval range, we estimate the tangent point vertical resolution to be roughly 171 m at 13.5 km and roughly 107 m at 18 km. We perform the retrieval on a 250 m sampling grid that is larger than this

uncertainty and we allow the regularization to control the vertical resolution. A small amount of damping is also applied to constrain the retrieval.

An example retrieval that was obtained by averaging the 10 successive coincident measurements is shown in Figure 18 (a). The figure shows 15 iterations; however, the retrieval converges in approximately 10 iterations. At each iteration, we produce a high-resolution forward modelled radiance for each of the 10 viewing geometries. Averaging is performed on the georeferenced forward

modelled images and the georeferenced measurement images using the exact same procedure. The associated averaging Kernel is shown in Figure 18 (b). Very little smoothing has been applied; therefore, the approximate vertical resolution of the measurements, determined from the full width half maximum of the averaging Kernel, is closely matched to the 250 m spacing of the retrieval grid. The measurement error due to Poisson noise is less than ± 1ppm for all altitudes.

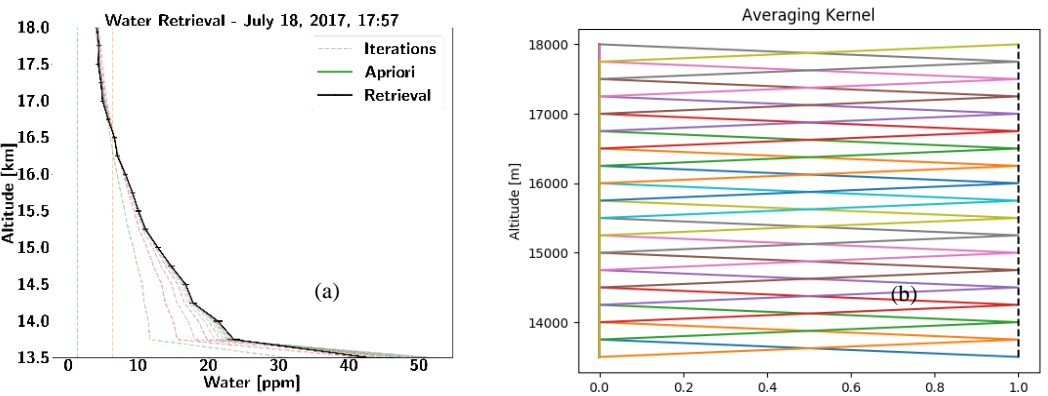

**Figure 18 Example SHOW water vapour retrieval (a) and the associated averaging Kernel (b).**

The retrieved water vapour profile is plotted along with the measured in-situ water vapour profile in Figure 19 (a). Since the in-situ profile is measured on a finer grid than the retrieval, the radiosonde data is transformed to the 250 m retrieval grid using the

method that is suggested by (Rodgers, 2000). In effect, this appropriately smooths the radiosonde measurements down to the vertical resolution of the retrieval grid. There is good agreement between the overall shape of the retrieved water vapour profile (shown in red) and the smoothed in-situ water profile (shown in blue) with the best agreement occurring at the high altitudes. The





ppm difference between the radiosonde and SHOW measurement is shown in Figure 19 (b). At higher altitudes, the difference is on the order of 1-2 ppm and the two measurements overlap within the error bars. However, it is clear that the SHOW measurements do not capture some of the small-scale variability that is observed with the radiosonde below 16 km where the difference is between 2-5 ppm. This type of difference is not uncommon when comparing limb sounding water vapour measurements with radiosondes

(Stiller et al., 2009).

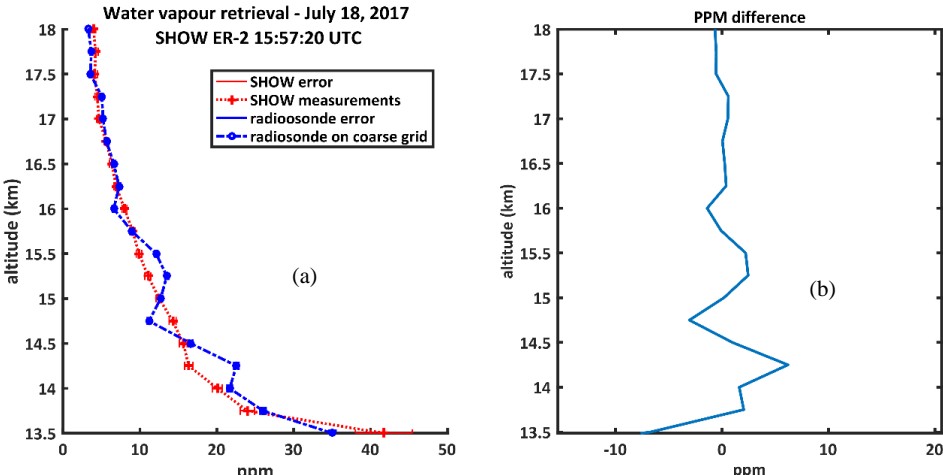

**Figure 19** SHOW water vapour retrieval plotted with the radiosonde measurements transformed to the coarse retrieval grid (a) and the ppm difference between the radiosonde and SHOW measurements.

There are several potential reasons for differences between the Vasaila RS41 radiosonde and SHOW. The most likely is differences in the viewing geometry between the two instruments, coupled with the fact that both instruments have unknown accuracies. The radiosonde samples the spatial and temporal variability of the water vapour field as a function of altitude, whereas SHOW measures the 1-dimensional water vapour abundance, which is heavily weighted to the tangent point. It is known that radiosonde accuracies

vary as a function of relative humidity and temperature and from sensor to sensor (Miloshevich et al., 2006). The Vaisala model RS41, utilized in this paper, has been shown to have improved accuracy over earlier models; however, errors between sensors on the order of 2-5 % have been recorded (Jensen et al., 2016). On the other hand, incomplete knowledge regarding the true instrument configuration (see Section 5) and the lack of knowledge of the true state of the atmosphere limits our ability to perform a more detailed characterization of the systematic effects associated with aliasing. In addition, error in the pointing knowledge of SHOW

measurements leads to a smearing of vertical information that is difficult to characterize when coupled with the aliasing effect. Therefore, more work is required to fully examine the expected biases between these two types of measurement.

**13      Conclusion**

The SHOW prototype instrument has performed successful demonstration flights from NASA's ER-2 airplane. In this paper, we have presented the characterization and the Level 0 to Level 1 processing of flight measurements that were obtained with SHOW

during an engineering flight that was performed on July 18th, 2017. Extracting water vapour profiles from the SHOW



measurements required a significant amount of calibration and characterization work and the development of an instrument model that is optimized to capture systematic variations that are observed in the measured spectra.

We have applied this approach to the SHOW measurements that were obtained during a period of stable flight where the SHOW instrument observed the same approximate column of air as a radiosonde that was launched from a JPL facility nearby Table
5    Mountain. These coincident measurements were compared to the SHOW measurements and were found to agree to within 1-5 ppm from 13.5 km – 18 km. The work presented in this paper provides initial validation of the SHOW measurement technique and demonstrates that high vertical resolution ($< 500$ m) measurements with $< \pm 1$ppm accuracy are feasible using this approach.

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
