# Peer review of "Spatial Heterodyne Observations of Water (SHOW) from a high altitude airplane: Characterization, performance and first results"

_Atmospheric Measurement Techniques, 2018_

## Referee Comment (RC1) · Anonymous Referee #1 · 27 Aug 2018

**GENERAL COMMENTS**

This nice paper describes the level 0 and level 1 processing scheme for a new limb-sounding heterodyne spectrometer called SHOW. It proceeds to describe the first test campaign, where the instrument worked well enough such that first water vapour retrieval results could be gained that compare favourable to co-located radiosonde measurements.

The paper is well written and its scope fits very well into AMT. It describes the individual steps of the processing in a very approachable manner (commendable!), but leaves often small but important details unmentioned or implicit. A minor textual revision that

ensures that all algorithms have been described with more rigour could turn this good paper into a great one. Specific notes below have been made below for the most important issues.

Further, there are several comments listed below, which must be addressed before I can recommend publication. I do not foresee any problem on the author's side to do so easily.

**SPECIFIC COMMENTS**

*page 23, Section 11.2:*
I find the use of the term "degraded vertical resolution" with respect to the pointing uncertainty misleading. The vertical resolution of the measurements is primarily determined by the FOV/PSF of the instrument. In case that the line-of-sight cannot be actively managed, aircraft movements during a measurement also worsen the vertical resolution. Another factor in determining the vertical resolution of the final product (compared to the measurement) is also a potentially employed regularisation.

However, here these factor are intermingled with an uncertainty in absolute pointing, which seemingly stems from inaccurate pitch readings by the ER-2 attitude system. Further factors may also contribute here, e.g. thermal contraction of the instrument. Such a bias in the pointing would be assumed to be constant during the interferogram acquisition. Such a bias would generally cause a shift in derived features (in contrast to a smearing out) as well as likely quantitative errors due to the wrong assignment of measurements to tangent altitudes and corresponding problems with pressure and temperature assumptions.

From the variance in radiation, you properly estimate an estimate for the precision of the employed line-of-sight angle, which shouldn't enter the vertical resolution. In contrast, I miss an estimate for the accuracy of the line-of-sight angle, which obviously can be calibrated on ground, but may change slightly upon mounting the instrument or may change due to thermal effects. Also, can the wings move in relation to the

main body of the aircraft depending on speed and prevalent winds? I also miss an estimate for the intra-interferogram variation of the pitch angle, which might be derived from highly-resolved aircraft pitch measurements that would, in fact, worsen the vertical resolution of the measurement.

*page 25, Section 11.3*
The section does a good job of giving a good overview of the employed technique but skimps on the detail. I assume that the computed correction factor for each spectral sample is of a multiplicative nature. While the method obviously works, it remains a question of how well this models the underlying failure and what error remains in the final data for proper error analysis.

The error may likely be assumed to be multiplicative in interferogram space due to the applied flat-fielding with potentially remaining higher-order effects (see Sect. 6). Such a multiplicative error in an interferogram sample will cause a sinusoidal error in the resulting spectrum. Due to the nature of the multiplicative error, I'd expect the influence of the pixels close to the ZOPD to be higher causing the error pattern to be of a more low-frequency nature and thus coloured noise.

So, I am looking for a physical motivation to the correction technique. If the effect is stable during this window, a point-wise multiplication in interferogram space should solve the issue, which would correspond to a convolution in spectral space.

Please motivate the correction technique and describe its application better to make clear its nature and implementation.

*page 30, lines 1ff:*
The agreement between the radiosonde and the SHOW retrieval results is indeed re-markable. However, the error bars due to noise errors are not applicable here, as noise is likely one of the smallest error sources of your retrieval compared to many of the instrumental uncertainties you (partially) corrected for and other inaccuracies by the forward model (e.g, line strength uncertainties, single/multiple-scattering) or other

influencing factors (uncertainties in pressure, assumed aerosol profile, solar radiation variation, horizontal homogeneities along the line-of-sight, . . . ).

The actual extent of the error bar is quite difficult to see on the plot. A logarithmic plot might be helpful here.

It is certainly not expected that the SHOW retrieval results agree with any other measurement within error bars induced by noise alone. The retrieval paper (Langille 2018) has, e.g., a sensitivity towards aerosol of 1 ppm, which also fully explains discrepancies at higher altitudes. For lower altitudes, the larger variability of water vapour in the stratosphere may, as rightly noted, explain observed discrepancies.

The paper should take systematic errors identified by Langille (2018) into account for the comparison. The plots should make identifying the errors better.

**MINOR REMARKS**

*page 1, lines 33ff:*
Amongst others, especially ACE-FTS has also significantly contributed to our knowledge about water vapour in the UTLS, partially due to its significantly higher resolution compared to the listed instruments. See for example doi:10.1029/2008JD009984. In contrast, it lacks in temporal and spatial coverage, obviously.

*page 6, lines 5:*
Reference of (Harlander, 1990) is not defined.

*page 6, lines 32:*
Are the ranges of the integral correct? With the variable transformation from sigma to kappa, I'd expect a corresponding change of integration range. In addition, I suspect also a missing factor. This makes only sense under the assumption of an appropriately chosen passband filter, which is only introduced later.

*page 9, line 24:*
Why is a nearest neighbour interpolation and not a linear one between the upper and

lower neighbour used? Especially in case of only a few bad pixels, this should reduce the error introduced due to the increase of radiation with decreasing tangent point altitude.

*page 9, lines 30ff:*
It is not clear what correction is performed on the interferograms with respect to the changing aircraft pitch. I interpret it in the way that not the interferogram is changed, but that each interferogram is associated with a tangent point altitude determined from the current aircraft pitch.

*page 9, lines 30ff:*
In what way is a possible bias between the LOS of SHOW and the given pitch angle of ER-2 determined? While the pitch variations of ER-2 can be readily applied, any systematic bias needs to be ruled out to properly assess the quality of the tangent altitudes.

*page 10, lines 4ff:*
It is unclear, what correction is described here. Section 11.2 describes a means of estimating remaining uncertainties in pitch/LOS after taking ER-2 pitch data into account, but does not describe correcting these. Different effects that are difficult to differentiate may cause the variation in radiation (changing scenery/clouds as well as inaccurate or imprecise pitch values from ER-2) and may a correction in contrast to an error estimation difficult.

*page 10, lines 19:*
For a non-linear retrieval, the matrix K should be marked with an index i or as being dependend of the current iterate $x_i$ as $K(x_i)$ or $F'(x_i)$. with $F'$ being the Jacobian of $F$.

*page 12, lines 25ff:*
From the description, it is unclear what uniformity or non-linearity correction is performed. The sentence that no uniformity correction is done contradicts a statement of

page 8 line 23.

*page 17, lines 3ff:*
One may also estimate the noise from the imaginary part of the spectrum after phase-correction. You typically operate on magnitude spectra. Could one still compute the phase from calibration measurements and use this to both reduce and estimate the noise?

*page 27, lines 8f:*
It is never spelled out precisely, how the correction is applied. It seems to be a multi-plicative factor?

*page 27, line 17f:*
There is a difference in scales between the two panels of Fig. 16. As such it is not apparent that the effect is minimal in wavenumber space. A delta plot would show that better in both instances. Especially the absorption feature at 1364.25 seems affected, but the l.h.s. might be affected by aliasing more. However also at 1365 is a visible different place with a large correction.

*page 29, line 10f:*
As shown in Fig. 18, the regularisation is insignificant and doesn't control or seemingly even influence the resolution.

*page 29, line 10:*
It is unclear what "damping" refers to and how it relates to the regularisation. One typically uses a trust-region method like Levenberg-Marquardt which contains something like a dampening factor. Here it sounds like the dampening would affect the result of the retrieval, which weren't the case for Levenberg-Marquardt.

**TECHNICAL CORRECTIONS**

*page 10, line 14:*
The cite of Rodgers should be "described by Rodgers (2000)".

*page 10, line 19ff:*
The used mathematical notation is formatted differently in the equation as in the main text (cursive vs. regular and probably in a different font).

*page 17, Figure 7:*
Especially the "(c)" in image (c) is barely readable. Potentially a white rectangle behind the letter or a "path effect" of matplotlib would help here.

*page 18, Equation (9):*
There are two consecutive "+" in the formula.

*page 24, Figure 13:*
Especially the "(b)" in image (b) is barely readable.

---

## Referee Comment (RC2) · Anonymous Referee #3 · 6 Sep 2018

This paper describes the details of measuring and retrieving water vapor from the SHOW instrument on a high altitude aircraft, including some first results compared to a nearby radiosonde profile. The development of the SHOW instrument is an exciting capability for the research community. Most of the paper involves characterization of the instrument and measurement details on the aircraft and in the laboratory. Overall, the paper provides a comprehensive explanation of these aspects, although I am not an expert on any of these details and cannot provide critical comments. Hopefully these will come from other reviewers. The comparison of the SHOW retrieval and the single radiosonde water vapor profile (Fig. 19) looks quite reasonable, although I question why the uncertainties are so small in both measurements (see below). Overall the

paper is clearly organized and well written, the figures are reasonable, and the topic is appropriate for AMT. I only have a few minor comments to contribute:

1) The uncertainties in Fig. 19 seem small to me. The radiosonde measurements (Vaisala RS41) probably makes accurate measurements (to a few %) in the upper troposphere down to 10-20 ppmv, but there are larger uncertainties for lower $H_2O$ amounts at higher altitudes. Where do the uncertainties (error bars) shown in Fig. 11 come from? The 1 ppmv uncertainty quoted here may be on the small side at upper levels. Likewise, the uncertainties in the SHOW retrieval look remarkably small, given all of the uncertainties and corrections discussed in the paper. I have to say this is a relatively minor point, given the quite good agreement in Fig. 19.

2) Some definitions need to be included: N (page 8, line -4 and elsewhere), and DN, PRNU (p. 11, Table 2 and elsewhere).

3) Figure 9a reproduces quite poorly, and no need to have a black background.

---

## Referee Comment (RC3) · Anonymous Referee #2 · 7 Sep 2018

**General Comments**

The manuscript details water vapor measurements made with a Spatial Heterodyne Spectrometer (SHS) from an ER-2 high altitude aircraft and compares retrieved water vapor densities with coincident radiosonde measurements. My overall impression of the manuscript is quite positive. It provides sufficient context and the detailed descriptions provided make it quite easy to follow. The agreement between the SHOW and the radiosonde water vapor densities is remarkable, particularly given the complexity of the L0 to L1 processing required in the analysis of the SHOW data. Given the quality of the manuscript, my specific and minor comments below are to be considered as advisory

[Figure]

to the authors and not requiring another detailed review.

Specific Comments

The manuscript details a series of steps required to remove instrumental effects from the level-0 data. The most complex of these stems from the combination of aliasing due to the passband of the interference filter spanning the Littrow wavenumber and a vertical fringe frequency tilt of the interferometer. The result is a row-dependent spectral modulation seen near left edges of Figures 7c and 7d and require that a detailed instrument model be in integral part of the retrieval. Although it's implicit in the discussion, I think it is important to point out that the difficulties and uncertainties associated with correcting for this effect could be eliminated by using a filter with passband shifted slightly to the red that blocks all light at wavelengths on the opposite side of Littrow. To simplify the analysis is there a plan to replace the interference filter for future flights of the instrument?

There are numerous places in the processing where a fitted high-order polynomial is subtracted from the data in an effort to assess the noise from the residual. It would be helpful to indicate the order of the polynomial used. Clearly with a very high order polynomial fit, some of the noise will be fit and the noise in the residual will be underestimated while fitting with a polynomial of too low an order will result in signal in the residual. How was the decision as to the order of polynomial used made?

Although not ultimately used in the analysis due to optical depth issues I found the description of the cloud artifacts evident in the lowest rows of Figure 12a and 12b somewhat lacking. If the entrance optics are anamorphic and aligned properly, they should completely defocus spatial information in the horizontal direction. Why then does the modulation in rows 0 to ∼30 in Figure 12a tilted? Could this possibly be a fringe due to a spectral line very close to the Littrow wavenumber?

It appears that amplitude spectra where used throughout in the analyses. If proper phase correction is performed the spectral information can be isolated to the real part
of the Fourier Transform only thereby reducing the shot noise contribution from the imaginary part and reducing the shot noise contribution by roughly the square root of two. That said, I suspect that the dominant source of uncertainty is not shot noise so perhaps this improvement is not worth the substantial effort required.

Connected to the previous comment, the error bars shown for the SHOW measurement in Figure 19 appear to be quite small. Do these error bars include the systematic effects associated with uncertainties in the retrieval or are they simply an indication of the photon shot noise component of the noise? It would most illuminating, if the systematic uncertainties could be quantified.

In general, I would suggest making a clearer distinction between statistical sources of noise (photo, dark, CCD read noise, etc.) and systematic sources of uncertainty (errors in the uniformity correction, uncertainties in the retrieval parameters, etc.). In various places in the manuscript all of these effects are referred to as "noise".

To understand the instrument, it would be helpful to add a short section and perhaps a figure describing the anamorphic optics that feed the interferometer and the optics between the interferometer and the detector.

Technical Corrections

Section 5 second paragraph, second sentence: Missing "to"

Section 6 third paragraph: N in the SNR equations should be defined

Equation 4: Define Ij and $\sigma$j.

Section 7, second sentence: Figure 3 reference should be Figure 2

Section 9.4, end of first paragraph: LightMachinery should be more completely referenced (e.g. LightMachinery, Inc., Ottawa, Ont).

Figure 9a: The black background makes it very difficult to read

Section 10, second paragraph, second sentence. "quiet" should be "quite"

Section 10, third paragraph: Acronym AFRC should be expanded

---

## Author Comment (AC1) · 20 Nov 2018

**Author response to Anonymous Referee #1**

First, we would like to thank you for your kind words regarding the content and structure of the manuscript. We understand that more details would have been helpful in some cases and we have made an attempt to include revisions to the manuscript that directly respond to your comments and concerns. Our responses to your comments are highlighted in red in the text below.

**Anonymous Referee #1**

**GENERAL COMMENTS**

This nice paper describes the level 0 and level 1 processing scheme for a new limb-sounding heterodyne spectrometer called SHOW. It proceeds to describe the first test campaign, where the instrument worked well enough such that first water vapour retrieval results could be gained that compare favourable to co-located radiosonde measurements.

The paper is well written and its scope fits very well into AMT. It describes the individual steps of the processing in a very approachable manner (commendable!), but leaves often small but important details unmentioned or implicit. A minor textual revision that ensures that all algorithms have been described with more rigour could turn this good paper into a great one. Specific notes below have been made below for the most important issues.

Further, there are several comments listed below, which must be addressed before I can recommend publication. I do not foresee any problem on the author's side to do so easily.

**SPECIFIC COMMENTS page 23, Section 11.2:**

I find the use of the term "degraded vertical resolution" with respect to the pointing uncertainty misleading. The vertical resolution of the measurements is primarily determined by the FOV/PSF of the instrument. In case that the line-of-sight cannot be actively managed, aircraft movements during a measurement also worsen the vertical resolution. Another factor in determining the vertical resolution of the final product (compared to the measurement) is also a potentially employed regularisation.

However, here these factor are intermingled with an uncertainty in absolute pointing, which seemingly stems from inaccurate pitch readings by the ER-2 attitude system. Further factors may also contribute here, e.g. thermal contraction of the instrument. Such a bias in the pointing would be assumed to be constant during the interferogram acquisition. Such a bias would generally cause a shift in derived features (in contrast to a smearing out) as well as likely quantitative errors due to the wrong assignment of measurements to tangent altitudes and corresponding problems with pressure and temperature assumptions.

From the variance in radiation, you properly estimate an estimate for the precision of the employed line-of-sight angle, which shouldn't enter the vertical resolution. In contrast, I miss an estimate for the accuracy of the line-of-sight angle, which obviously can be calibrated on ground, but may change slightly upon mounting the instrument or may change due to thermal effects. Also, can the wings move in relation to the main body of the aircraft depending on speed and prevalent winds? I also miss an

estimate for the intra-interferogram variation of the pitch angle, which might be derived from highly-resolved aircraft pitch measurements that would, in fact, worsen the vertical resolution of the measurement.

At the beginning of the SHOW ER-2 project and leading up to the ER-2 demonstration flights, there were several concerns regarding the stability of the viewing geometry due to the aircraft motions. These concerns were primarily focused on potential vibrations in the aircraft wings (through flexing) or inherent pitch variations in the aircraft body.   Clearly, mapping the observed lines of sight in the images to the corresponding georeferenced tangent altitudes requires us to account for these variations.  After the demonstration fights, the concerns regarding the aircraft pitch variations were partially alleviated since we found that the aircraft attitude information (obtained from a sensor located in the fuselage) could be used in combination with the atmospheric signals to obtain georeferenced images. This is the mapping that is described in Section 11.2.   On the other hand, we found it difficult to quantify any possible "flexing" of the wings during the flight since the attitude sensor is located in the fusealage of the airplane. Quantifying this effect would require the use of a sensor in the wingpod to monitor the absolute pointing.  Indeed, future versions of the instrument should make use of such a configuration to remove this uncertainty.

We agree that the use of the term "degraded vertical resolution" with respect to the pointing accuracy is not entirely appropriate. In order to address your concern, we have removed this wording and we have revised small parts of the manuscript to include a description of the uncertainty in the absolute pointing which was confirmed on the ground to 0.01 degrees. We separate this uncertainty from the instantaneous field of view (or PSF) of the instrument which sets the maximum possible vertical resolution that is possible for the measurements.

The vertical resolution of the SHOW measurements is now discussed in the following manner in the manuscript:

1.  In section 9.3.2, we present the vertical field of view measurement that was performed on the ground in the lab at ABB Ltd.  This provides us with a mapping of angle of incidence (or line of sight angle) to row number (Figure 4) and an estimate of the instantaneous field of view of the instrument (or point spread function).  The instantaneous field of view of the instrument was measured to be roughly 0.0176 degrees.  This sets the maximum vertical resolution at the limb that can be obtained with the instrument.

2.  Aircraft movements that occur during the acquisition of an interferogram will also degrade the vertical resolution. This effect is negligible for the SHOW measurements, since the frame rate is high enough that the aircraft motions can be considered stable over a single acquisition(Figure 13 (a)).

3.  In practice, the spacing of the retrieval grid and the type of and amount regularization also impacts the vertical resolution. In our case, very little smoothing is applied and the regularization has very little impact on the retrieved water vapour profile. This is illustrated by the averaging kernel in Figure 18 (b) which has widths that are equal to the 250 m retrieval grid.

The absolute pointing is now discussed in the manuscript with the following content included to address your concerns:

1. The SHOW instrument is mounted in the wing pod of the ER-2 using a mounting frame that is designed to mechanical tolerances by ABB to point the 4 degree field of view of the instrument downward by 3.23 degrees. Prior to the ER-2 flight, the pointing of the SHOW instrument was measured in the hangar at AFRC using a point source, laser levels and tape measures. Using this approach, the accuracy of the pointing is estimated to be roughly ±0.01 degrees. This uncertainty is now included in the discussion of Section 11.2.

2. The aircraft wings can flex during flight and this flex is not captured by the aircraft attitude information. This introduces a source of uncharacterized uncertainty in the absolute pointing of the instrument; however, good agreement between comparisons of the radiance profiles from the forward model and the SHOW measurements indicate that this effect is likely small.

3. We find it possible, but unlikely that the thermal contraction of the instrument could contribute to variations in the absolute pointing. The instrument is mounted inside the wing pod with a thick aluminum frame that is not expected to contract significantly given the observed temperature variations during the flight.

4. In Section 11.2, we utilize the aircraft pitch information to map the interferogram rows to the lines of sight at the limb. The precision of this mapping is estimated by quantifying the remaining variability in an observed cloud feature in the column averaged interferogram images ( Figure 13). The remaining variability amounts to an uncertainty in the absolute pointing of the instrument of ~0.032 degrees. This uncertainty is no longer lumped in with the discussion of the vertical resolution of the measurements.

**Page 25, Section 11.3**

The section does a good job of giving a good overview of the employed technique but skimps on the detail. I assume that the computed correction factor for each spectral sample is of a multiplicative nature. While the method obviously works, it remains a question of how well this models the underlying failure and what error remains in the final data for proper error analysis.

The error may likely be assumed to be multiplicative in interferogram space due to the applied flat-fielding with potentially remaining higher-order effects (see Sect. 6). Such a multiplicative error in an interferogram sample will cause a sinusoidal error in the resulting spectrum. Due to the nature of the multiplicative error, I'd expect the influence of the pixels close to the ZOPD to be higher causing the error pattern to be of a more low-frequency nature and thus coloured noise.

So, I am looking for a physical motivation to the correction technique. If the effect is stable during this window, a point-wise multiplication in interferogram space should solve the issue, which would correspond to a convolution in spectral space.

Please motivate the correction technique and describe its application better to make clear its nature and implementation.

In Section 6, we describe the various sources of noise in the interferogram samples that are obtained with the SHS. The primary sources of noise are due to photon counting and the residual inter-sample

variability due to a less-than-perfect flat field correction.  This flat field correction included variations due to optical effects as well as variations in the response of the pixels across the detector.

The SNR of the individual measurements is not an issue for the SHOW instrument – the SNR is high enough to minimize the impact of the noise associated with photon counting. For the SHOW measurements, we found that the primary source of noise in the interferogram samples is the inter-sample variability due to a less than ideal flat-field.

Indeed, the combined flat field correction and non-uniformity correction is multiplicative; however, the residual variation that remains after the application can be viewed as a perturbation.  As noted in Section 6, this underlying failure leads to an effectively random relative pixel variation $\delta S/S$ across the interferogram row where S is the signal at the interferogram peak and $\delta S$ is the error.   In this case, the noise level on particular sample is not proportional to the source intensity and is fixed for a fixed source strength. It adds a small perturbation to each sample in a similar manner to digitization noise ( see Davis (2003)).

This feature is discussed by Englert et al. (2006) where he notes that such a variation results in an SNR in the spectral samples of between $\sqrt{N/2}\, S/\delta S$ and $\sqrt{2/N}\, S/\delta S$ due to multiplex noise propagation for the case of a monochromatic line or continuum respectively. SHOW utilizes a relatively broad spectral feature and therefore suffers a loss in SNR closer to the continuum case.

While the pattern is effectively random over the several hundred interferogram samples, the inter-sample variability remains fixed for a fixed input intensity. If we take the same measurement hundreds of time with a stable source of fixed intensity, then we find that the amplitudes of inter-sample variations remains fixed in the interferograms, as well as, the spectra.   The effect be viewed as adding and subtracting a fixed set of very small perturbations to the interferogram samples.  The FFT of the resulting interferogram gives the desired spectrum plus a fixed (in wavenumber space) set of associated perturbations. Since the perturbations are small (< 5 %), a multiplicative factor can be isolated using the approach discussed in Section 11.3. The text in Section 6 and in Section 11.3 has been updated to make this physical understanding more clear.

**Page 30, lines 1ff:**

The agreement between the radiosonde and the SHOW retrieval results is indeed remarkable. However, the error bars due to noise errors are not applicable here, as noise is likely one of the smallest error sources of your retrieval compared to many of the instrumental uncertainties you (partially) corrected for and other inaccuracies by the forward model (e.g, line strength uncertainties, single/multiple-scattering) or other influencing factors (uncertainties in pressure, assumed aerosol profile, solar radiation variation, horizontal homogeneities along the line-of-sight, . . . ).

 The actual extent of the error bar is quite difficult to see on the plot. A logarithmic plot might be helpful here. It is certainly not expected that the SHOW retrieval results agree with any other measurement within error bars induced by noise alone. The retrieval paper (Langille 2018) has, e.g., a sensitivity towards aerosol of 1 ppm, which also fully explains discrepancies at higher altitudes. For lower altitudes, the larger variability of water vapour in the stratosphere may, as rightly noted, explain observed discrepancies.

The paper should take systematic errors identified by Langille (2018) into account for the comparison. The plots should make identifying the errors better.

Agreed, the main source of error is not associated with measurement noise and the error bars will be removed from the plot and text has been added to the Figure caption and in the text of the manuscript to explicitly discuss the uncertainties. The manuscript has been edited to relate the discrepancies in the plot back to the sensitivity study (Langille, et al., 2018).

**MINOR REMARKS page 1, lines 33ff:**

Amongst others, especially ACE-FTS has also significantly contributed to our knowledge about water vapour in the UTLS, partially due to its significantly higher resolution compared to the listed instruments. See for example doi:10.1029/2008JD009984. In contrast, it lacks in temporal and spatial coverage, obviously.

A reference to an ACP ACE papers has now been added to this paragraph.

page 6, lines 5: Reference of (Harlander, 1990) is not defined.

This has been changed to (Harlander, 1991) and (Harlander, 1992).

page 6, lines 32: Are the ranges of the integral correct? With the variable transformation from sigma to kappa, I'd expect a corresponding change of integration range. In addition, I suspect also a missing factor. This makes only sense under the assumption of an appropriately chosen passband filter, which is only introduced later.

You are correct. The range needed to be updated to reflect the change in variables. Thank you for picking up on that one.

page 9, line 24: Why is a nearest neighbour interpolation and not a linear one between the upper and lower neighbour used? Especially in case of only a few bad pixels, this should reduce the error introduced due to the increase of radiation with decreasing tangent point altitude.

For this iteration of the SHOW instrument we chose nearest neighbor interpolation since it appeared to to work reasonably well in the processing of lab measurements. We are in the fortunate scenario where the detector has very few bad pixels and in most cases, a single row has none or only one bad pixel. In this case, the nearest neighbor interpolation works fine. In future campaigns we plan to investigate other approaches to this procedure.

page 9, lines 30ff: It is not clear what correction is performed on the interferograms with respect to the changing aircraft pitch. I interpret it in the way that not the interferogram is changed, but that each interferogram is associated with a tangent point altitude determined from the current aircraft pitch.

In fact, this is not a correction, but rather a procedure that is applied to map the interferogram rows to tangent altitude at the limb. The manuscript has been edited to make this more clear.

page 9, lines 30ff: In what way is a possible bias between the LOS of SHOW and the given pitch angle of ER-2 determined? While the pitch variations of ER-2 can be readily applied, any systematic bias needs to be ruled out to properly assess the quality of the tangent altitudes.

There is the very real possibility of a systematic bias between the LOS of SHOW and the pitch angle of ER-2. As mentioned in the response to your "Specific Comment", the bias between the SHOW LOS and the ER-2 pitch was measured in the hangar to an accuracy of ±0.01 degrees. The manuscript has been updated to note this pointing accuracy.

page 10, lines 4ff: It is unclear, what correction is described here. Section 11.2 describes a means of estimating remaining uncertainties in pitch/LOS after taking ER-2 pitch data into account, but does not describe correcting these. Different effects that are difficult to differentiate may cause the variation in radiation (changing scenery/clouds as well as inaccurate or imprecise pitch values from ER-2) and may a correction in contrast to an error estimation difficult.

This is not really a correction but a mapping that maps interferogram rows to tangent altitude. The remaining variability is quantified by determining the variability on what appears to be a cloud feature. The text has been edited to make this clear. The remaining variability amounts to an uncertainty in the absolute pointing of the instrument of ±0.032 degrees.

page 10, lines 19: For a non-linear retrieval, the matrix K should be marked with an index i or as being dependent of the current iterate xi as K(xi) or F 0 (xi). with F 0 being the Jacobian of F.

This has been updated in the text.

page 12, lines 25ff: From the description, it is unclear what uniformity or non-linearity correction is performed. The sentence that no uniformity correction is done contradicts a statement of page 8 line 23.

The non-uniformity correction is performed as part of the flat-field. The text has been edited to remove the contradictory statement.

page 17, lines 3ff: One may also estimate the noise from the imaginary part of the spectrum after phasecorrection. You typically operate on magnitude spectra. Could one still compute the phase from calibration measurements and use this to both reduce and estimate the noise?

This type of correction is not feasible with the current configuration of the SHOW instrument due to the presence of the aliasing effect. As mentioned in Section 5, aliasing leads to a complicated phase relationship that is extremely difficult to isolate. This relationship changes with the

page 27, lines 8f: It is never spelled out precisely, how the correction is applied. It seems to be a multiplicative factor?

Yes, the correction factor is a multiplicative factor. The procedure that is used to isolate this correction and apply it to the data has been made more clear in the manuscript. Specific sections that have been edited are Section 6 and Section 11.3.

page 27, line 17f: There is a difference in scales between the two panels of Fig. 16. As such it is not apparent that the effect is minimal in wavenumber space. A delta plot would show that better in both instances. Especially the absorption feature at 1364.25 seems affected, but the l.h.s. might be affected by aliasing more. However also at 1365 is a visible different place with a large correction.

In this Figure, we were attempting to point out that the primary variability is in the vertical dimension. The statement saying the effect in wavenumber space is minimal has been removed.

page 29, line 10f: As shown in Fig. 18, the regularisation is insignificant and doesn't control or seemingly even influence the resolution.

This section has been edited to make it clear that the employed regularization does not significantly impact the vertical resolution.

page 29, line 10: It is unclear what "damping" refers to and how it relates to the regularisation. One typically uses a trust-region method like Levenberg-Marquardt which contains something like a dampening factor. Here it sounds like the dampening would affect the result of the retrieval, which weren't the case for Levenberg-Marquardt.

The damping term here does not influence the retrieval in the case that the retrieval has converged. The wording has been changed to state that "A small amount of damping is applied to ensure convergence of the retrieval"

**TECHNICAL CORRECTIONS**

page 10, line 14: The cite of Rodgers should be "described by Rodgers (2000)".

This has been edited in the text

page 10, line 19ff: The used mathematical notation is formatted differently in the equation as in the main text (cursive vs. regular and probably in a different font).

This has been edited in the text

page 17, Figure 7: Especially the "(c)" in image (c) is barely readable. Potentially a white rectangle behind the letter or a "path effect" of matplotlib would help here.

This has been edited in the text

page 18, Equation (9): There are two consecutive "+" in the formula.

This has been edited

page 24, Figure 13: Especially the "(b)" in image (b) is barely readable

This has been edited

---

## Author Comment (AC2) · 20 Nov 2018

**Response to Anonymous Referee #3**

Thank you for your comments on the manuscript. Please find our responses to your comments below.

**Anonymous Referee #3**

This paper describes the details of measuring and retrieving water vapor from the SHOW instrument on a high altitude aircraft, including some first results compared to a nearby radiosonde profile. The development of the SHOW instrument is an exciting capability for the research community. Most of the paper involves characterization of the instrument and measurement details on the aircraft and in the laboratory. Overall, the paper provides a comprehensive explanation of these aspects, although I am not an expert on any of these details and cannot provide critical comments. Hopefully these will come from other reviewers. The comparison of the SHOW retrieval and the single radiosonde water vapor profile (Fig. 19) looks quite reasonable, although I question why the uncertainties are so small in both measurements (see below). Overall the paper is clearly organized and well written, the figures are reasonable, and the topic is appropriate for AMT. I only have a few minor comments to contribute:

1) The uncertainties in Fig. 19 seem small to me. The radiosonde measurements (Vaisala RS41) probably makes accurate measurements (to a few %) in the upper troposphere down to 10-20 ppmv, but there are larger uncertainties for lower H2O amounts at higher altitudes. Where do the uncertainties (error bars) shown in Fig. 11 come from? The 1 ppmv uncertainty quoted here may be on the small side at upper levels. Likewise, the uncertainties in the SHOW retrieval look remarkably small, given all of the uncertainties and corrections discussed in the paper. I have to say this is a relatively minor point, given the quite good agreement in Fig. 19.

The uncertainties in Figure 19 only include an estimate of the measurement uncertainty calculated using the specifications for the Vaisala RS41. The raw measurement uncertainty on any given radiosonde sample is on the order of 1-2 ppm; however, these uncertainties are further reduced since they are transformed to the lower vertical resolution retrieval grid of the SHOW measurements. The radiosonde error bars do not include any estimates of the accuracy of the measurement. In the manuscript we point out that the accuracy of the measurements is a function of relative humidity and the temperature and that these accuracies vary from sensor to sensor. Similarly, the error bars on the SHOW water vapour measurements only include measurement noise. In our case, the errors due to measurement noise are quite small due to the high SNR of the measurements.

We agree that the error bars are too small to be useful in the Figures. We have removed from the figure and the text has been edited to make it clear that these uncertainties are not the limiting errors in the measurements.

2) Some definitions need to be included: N (page 8, line -4 and elsewhere), and DN, PRNU (p. 11, Table 2 and elsewhere).

The text has been edited to provide definitions of these quantities.

3) Figure 9a reproduces quite poorly, and no need to have a black background.

This figure was provided by AFRC and it not available with another background.

---

## Author Comment (AC3) · 20 Nov 2018

**Response to Anonymous Referee #2**

Thank you very much for your comments on the manuscript.  Please find our responses to your comments below highlighted in red.

**Anonymous Referee #2**

**General Comments**

The manuscript details water vapor measurements made with a Spatial Heterodyne Spectrometer (SHS) from an ER-2 high altitude aircraft and compares retrieved water vapor densities with coincident radiosonde measurements. My overall impression of the manuscript is quite positive. It provides sufficient context and the detailed descriptions provided make it quite easy to follow. The agreement between the SHOW and the radiosonde water vapor densities is remarkable, particularly given the complexity of the L0 to L1 processing required in the analysis of the SHOW data. Given the quality of the manuscript, my specific and minor comments below are to be considered as advisory to the authors and not requiring another detailed review.

**Specific Comments**

The manuscript details a series of steps required to remove instrumental effects from the level-0 data. The most complex of these stems from the combination of aliasing due to the passband of the interference filter spanning the Littrow wavenumber and a vertical fringe frequency tilt of the interferometer. The result is a row-dependent spectral modulation seen near left edges of Figures 7c and 7d and require that a detailed instrument model be in integral part of the retrieval. Although it's implicit in the discussion, I think it is important to point out that the difficulties and uncertainties associated with correcting for this effect could be eliminated by using a filter with passband shifted slightly to the red that blocks all light at wavelengths on the opposite side of Littrow. To simplify the analysis is there a plan to replace the interference filter for future flights of the instrument?

You are correct. This effect can be eliminated by changing the filter. Indeed, after the first ER-2 demonstration flights we obtained the financial resources to procure a new filter that is shifted so the Littrow wavelength sits at the edge of the filter passband. This ensures that there is negligible aliasing from the opposite side of Littrow. The filter was procured and installed in the Spring of 2018. The new configuration has been characterized in the lab and shows no signs of the aliasing effect.  Several lines of text have been added to Section 5 of the manuscript to explicitly point out that this effect is entirely avoidable by design.

There are numerous places in the processing where a fitted high-order polynomial is subtracted from the data in an effort to assess the noise from the residual. It would be helpful to indicate the order of the polynomial used. Clearly with a very high order polynomial fit, some of the noise will be fit and the noise in the residual will be underestimated while fitting with a polynomial of too low an order will result in signal in the residual. How was the decision as to the order of polynomial used made?

In most cases, the high order polynomial was of order 8.  However, we did not do a detailed analysis of the impact of lower or higher orders.  For most of the cases, we are just using this approach to obtain a first order approximation of the noise in the measurements to check that photon-noise is the primary

measurement noise on an individual interferogram sample. The text has been edited to explicitly state the order of polynomial that was used.

Although not ultimately used in the analysis due to optical depth issues I found the description of the cloud artifacts evident in the lowest rows of Figure 12a and 12b somewhat lacking. If the entrance optics are anamorphic and aligned properly, they should completely defocus spatial information in the horizontal direction. Why then does the modulation in rows 0 to ~30 in Figure 12a tilted? Could this possibly be a fringe due to a spectral line very close to the Littrow wavenumber?

Thanks for pointing this out. After a second look at the images, it turns out that the primary effect is likely due to removal of the DC bias in the presence of nearly saturated pixels at the interferogram center burst. It is known that saturated pixels causes the pixels in the corresponding row to have  non-ideal behavior.  The manuscript has been edited to note that the cause of this effect is not completely clear; however, it is believed that the culprit is likely saturated pixels at the interferogram center burst combined with the DC bias removal. In any case, we do not use these rows in the retrieval and the effect does not appear in the rows above the cloud feature where the retrieval is performed.

It appears that amplitude spectra where used throughout in the analyses. If proper phase correction is performed the spectral information can be isolated to the real part of the Fourier Transform only thereby reducing the shot noise contribution from the imaginary part and reducing the shot noise contribution by roughly the square root of two. That said, I suspect that the dominant source of uncertainty is not shot noise so perhaps this improvement is not worth the substantial effort required.

Unfortunately, we could not perform a proper phase correction on the SHOW measurements due to the presence of aliasing in the system.  We also did not have access to a tunable laser that could be used to perform this characterization.  Such a characterization will be performed prior to the next SHOW measurement campaign using the new filter.

Connected to the previous comment, the error bars shown for the SHOW measurement in Figure 19 appear to be quite small. Do these error bars include the systematic effects associated with uncertainties in the retrieval or are they simply an indication of the photon shot noise component of the noise? It would most illuminating, if the systematic uncertainties could be quantified. In general, I would suggest making a clearer distinction between statistical sources of noise (photo, dark, CCD read noise, etc.) and systematic sources of uncertainty (errors in the uniformity correction, uncertainties in the retrieval parameters, etc.). In various places in the manuscript all of these effects are referred to as "noise".

The error bars shown in this figure only include measurement noise.  The error bars do not include systematic effects associated with uncertainties in the retrieval.   In our sensitivity study (Langille et al (2018)) we examined some the sensitivities of the retrieval assuming an ideal instrument configuration. In the current work, the primary source of uncertainty is our ability to accurately capture the systematics associated with aliasing.    We have made edits to the manuscript to make a clearer distinction between the various sources of noise.

To understand the instrument, it would be helpful to add a short section and perhaps a figure describing the anamorphic optics that feed the interferometer and the optics between the interferometer and the detector.

The anamporphic optics and imaging configuration are discussed in detail in the design paper (Langille et al., 2017).

**Technical Corrections**

Section 5 second paragraph, second sentence: Missing "to"

Corrected in the manuscript
Section 6 third paragraph: N in the SNR equations should be defined

Corrected in the manuscript

Equation 4: Define Ij and σj.

Corrected in the manuscript

Section 7, second sentence: Figure 3 reference should be Figure 2

Corrected in the manuscript

Section 9.4, end of first paragraph: LightMachinery should be more completely referenced (e.g. LightMachinery, Inc., Ottawa, Ont).

Corrected in the manuscript

Figure 9a: The black background makes it very difficult to read

Unfortunately, this is the only version of the flight path that was made available by AFRC.